# Future-Gain Guided Test-Time Learning for Large Language Models

**Langyu Bian** [1 *]   **Jinwu Hu** [1 2 *]   **Zitian Zhang** [1]   **Dongjin Yang** [1]   **Yufeng Wang** [1 3]
**Qing Du** [1 †]   **Qi Chen** [4 †]   **Mingkui Tan** [1 5 †]

## Abstract

Large language models (LLMs) inevitably encounter distribution shifts during real-world deployment, leading to performance degradation. Although test-time learning (TTL) adapts LLMs from unlabeled test streams, applying entropy minimization to autoregressive generation faces two challenges: *(i)* early decoding errors can steer later tokens off track, and updating on them can push the model further off course, and *(ii)* updates on unreliable tokens can amplify confident error predictions and trigger model collapse. To address these challenges, we propose **F**uture-**G**ain Guided **T**est-**T**ime **L**earning (**FG-TTL**) for LLMs, which learns selectively from the model's own generations. Our key idea is to update only on tokens that reduce uncertainty in subsequent generation rather than tokens that are merely uncertain at the current step. Specifically, we develop a **F**uture-Gain Guided **T**oken **S**election (**FTS**) strategy to decide where to learn. We introduce *Future-Gain* as a token-level metric for this purpose and update the model only on high-gain tokens, concentrating learning on informative positions and mitigating temporal error propagation. In addition, we design a **R**isk-**A**ware **A**daptation (**RAA**) mechanism that controls how strongly to update by combining gain-based weighting with adaptive temperature scaling based on intrinsic uncertainty, suppressing unreliable gradients while enabling stronger learning on high-gain tokens. Experiments on six benchmarks with three LLM backbones show that FG-TTL achieves the best average performance.

[*]Equal contribution [1]School of Software Engineering, South China University of Technology, China [2]Pazhou Laboratory, China [3]Peng Cheng Laboratory, China [4]Australian Institute for Machine Learning, Adelaide University [5]Key Laboratory of Big Data and Intelligent Robot, Ministry of Education, China. Correspondence to: Mingkui Tan <mingkuitan@scut.edu.cn>, Qing Du <duqing@scut.edu.cn>, Qi Chen <qi.chen04@adelaide.edu.au>.

*Proceedings of the 43$^{rd}$ International Conference on Machine Learning*, Seoul, South Korea. PMLR 306, 2026. Copyright 2026 by the author(s).

## 1. Introduction

Large language models (LLMs) have achieved remarkable success across a wide range of tasks, from complex logical reasoning (Wei et al., 2022b; OpenAI, 2023; DeepSeek-AI, 2025) to creative generation (Chen et al., 2024b; Wang et al., 2025a). This success is primarily due to extensive pre-training on vast amounts of data, which enables the models to capture rich linguistic representations (Team, 2024; Cao et al., 2024; Hu et al., 2026). However, in real-world deployment, models encounter continuous streams of data in which the target distribution invariably differs from the source distribution due to evolving user intent, domain-specific terminology, or temporal shifts (Akyürek et al., 2025; Hu et al., 2025). Consequently, static LLMs often degrade substantially under distribution shift, with failures such as hallucinations and incorrect reasoning (Zhang et al., 2025b).

To mitigate performance degradation in real-world deployment, prior efforts can be broadly grouped into three categories. *Fine-Tuning (FT)* (Hu et al., 2022; Thirunavukarasu et al., 2023; Yuan et al., 2025) updates model parameters using newly collected data, typically relying on task-specific supervision or human feedback. While effective when labeled data is available, acquiring such annotations is often costly and impractical at scale. *Test-Time Training (TTT)* (Hardt & Sun, 2024; Hübotter et al., 2025) further adapts the model during inference by fine-tuning on relevant examples retrieved from source datasets. However, TTT assumes access to proprietary training data or knowledge bases that are frequently unavailable in practice, and it incurs substantial retrieval overhead. In contrast, *Test-Time Learning (TTL)* (Hu et al., 2025; Zuo et al., 2025; Xu et al., 2025) leverages the unlabeled test inputs themselves to induce self-supervised signals and updates only a lightweight subset of parameters at test time, thereby avoiding explicit supervision and dependence on external resources.

Despite the progress of existing TTL methods, several critical issues remain. Some approaches, such as TLM (Hu et al., 2025), attempt to align LLMs to the target domain by minimizing the perplexity of incoming inputs. However, reducing perplexity on inputs mainly improves input domain fit, but it does not guarantee that the model will decode in the right way, such as following task-specific so-

lution patterns or maintaining coherent reasoning (Xu et al., 2025). This naturally motivates leveraging the model's own generations as self-supervision, so that test-time updates can more directly act on the decoding process. A closely related idea has been extensively studied in computer vision: entropy-minimization based test-time adaptation (TTA) (Wang et al., 2021; Niu et al., 2022; Zhang et al., 2025a), such as TENT (Wang et al., 2021) and EATA (Niu et al., 2022), updates a lightweight subset of parameters to encourage confident predictions on unlabeled test samples. However, directly transferring entropy minimization to autoregressive LLM generation is non-trivial, mainly due to two challenges. **(i)** Autoregressive decoding is temporally coupled, so early errors can cascade into later tokens and cause update drift (Li et al., 2026). **(ii)** Entropy minimization on unreliable tokens can amplify confident error predictions and even trigger model collapse (Niu et al., 2023).

To address these challenges, we propose **F**uture-**G**ain Guided **T**est-**T**ime **L**earning (**FG-TTL**) for large language models. Our key insight is that not all generated tokens are equally useful for online adaptation. Motivated by cognitive principles of uncertainty reduction (Bennett et al., 2016) and the information-theoretic notion of predictive information (Bialek et al., 2001), we focus on tokens that act as informational pivots and clarify the downstream decoding trajectory. Accordingly, a reliable update signal should be derived from tokens that reduce uncertainty in subsequent generation, rather than from tokens that are merely uncertain at the current step or simply have low instantaneous likelihood. Specifically, we develop a **F**uture-Gain Guided **T**oken **S**election (**FTS**) strategy to decide where to learn by prioritizing tokens that reduce downstream uncertainty in subsequent generations. We introduce *Future-Gain* as a token-level metric for this purpose and update the model only on high-gain tokens, concentrating learning on informative positions and mitigating temporal error propagation. In addition, we design a **R**isk-**A**ware **A**daptation (**RAA**) mechanism that controls how strongly to update. It modulates update strength through gain-based weighting and adaptive temperature scaling based on intrinsic uncertainty, which reduces gradient magnitude when the model is unreliable and enables stronger learning on high-gain tokens. Our main contributions are as follows:

- **Future-Gain Guided Token Selection:** We propose a Future-Gain based token selection strategy for test-time learning, which prioritizes tokens that reduce downstream uncertainty in generation. This enables selective updates on informative positions and mitigates temporal error propagation in autoregressive decoding.

- **Risk-Aware Adaptation:** We introduce a risk-aware update mechanism via gain-based weighting and adaptive temperature scaling to control the intensity of parameter updates. This suppresses unreliable gradients

and stabilizes entropy-based adaptation, reducing update drift and preventing model collapse.

- **Experimental Validation:** We evaluate FG-TTL on six reasoning benchmarks and DomainBench. FG-TTL achieves the best average EM among TTL and TTA baselines, improving the strongest baseline by 1.60% on Llama3.1-8B-Instruct, and achieves the best average Rouge-Lsum on DomainBench.

## 2. Related Work

**Fine-Tuning (FT)** updates the parameters of a pre-trained model using newly collected data, typically guided by task-specific supervision or human feedback (Thirunavukarasu et al., 2023; Yuan et al., 2025; Wei et al., 2022a; Ouyang et al., 2022; Hu et al., 2022; Dettmers et al., 2023; Wang et al., 2025c). To enhance capability across diverse tasks, methods such as FLAN (Wei et al., 2022a) utilize instruction tuning to enhance zero-shot generalization. Furthermore, InstructGPT (Ouyang et al., 2022) uses reinforcement learning from human feedback to align models with user intent, ensuring helpful and safe outputs. To address the high computational costs of such updates, LoRA (Hu et al., 2022) freezes pre-trained weights and optimizes injected low-rank matrices, achieving comparable performance with significantly reduced overhead. However, acquiring such high-quality supervision is often costly and impractical at scale, limiting the feasibility of fine-tuning for continuous adaptation in dynamic environments.

**Test-Time Training (TTT)** adapts models during inference by fine-tuning on relevant data retrieved from external knowledge bases or training sets (Hardt & Sun, 2024; Hübotter et al., 2025; Sun et al., 2026). For instance, TTT-NN (Hardt & Sun, 2024) retrieves documents to temporarily update the model, thereby enhancing performance without the computational cost of extending context length. Building on this, SIFT (Hübotter et al., 2025) improves the data quality through an active selection algorithm that prioritizes samples with high information gain, effectively reducing redundancy compared to standard nearest-neighbor retrieval. Despite their effectiveness, TTT incurs substantial retrieval costs and requires external data availability, which limits its utility in strictly source-free or isolated test-time settings.

**Test-Time Learning (TTL)** exploits unlabeled test data to derive self-supervised signals and updates only a small set of lightweight parameters during inference, eliminating the need for explicit supervision or external resources (Hu et al., 2025; Zuo et al., 2025; Xu et al., 2025; Wang et al., 2025b; Islah et al., 2025). TLM (Hu et al., 2025) focuses on minimizing input perplexity to capture target domain distributions. SyTTA (Xu et al., 2025) extends this by incorporating output predictive entropy to further stabilize adaptation. TTRL (Zuo et al., 2025) targets complex reasoning tasks

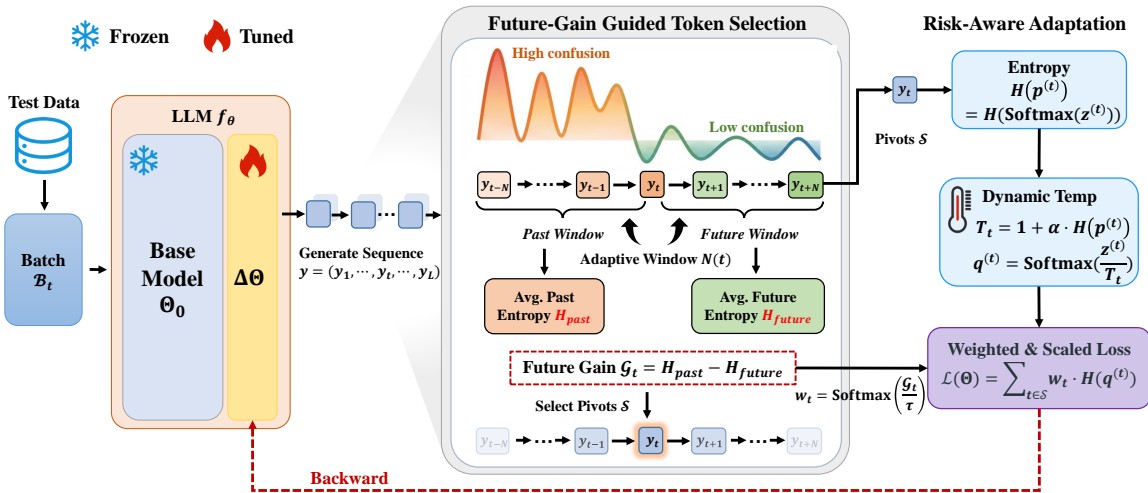

Figure 1. Overview of the proposed FG-TTL. For each incoming batch $\mathcal{B}_t$, the model first generates a response $y$. The **Future-Gain Guided Token Selection** strategy analyzes this sequence to identify pivotal tokens $\mathcal{S}$ by calculating their Future-Gain $\mathcal{G}_t$ using an adaptive window $N(t)$. Subsequently, the **Risk-Aware Adaptation** mechanism computes the final loss signal $\mathcal{L}$ by combining gain-based weights $w_t$ with an adaptive temperature scaling $T_t$ for each pivot. This loss is then back-propagated to update only the LoRA parameters.

by formulating intrinsic rewards from the inference process itself. However, these methods typically treat generated tokens indiscriminately or rely on static metrics, which risks reinforcing confident error predictions and triggering model collapse when the self-supervised signals are unreliable.

## 3. Problem Formulation

We consider a large language model (LLM) $f_\Theta$ parameterized by $\Theta$, initialized from a pre-trained state $\Theta_0$ that has been supervised fine-tuned (SFT) on a source dataset $\mathcal{D}_{\text{src}} \sim P(x, y)$. During the SFT stage, the model learns to approximate the conditional probability $p(y|x)$ by capturing linguistic patterns and reasoning heuristics inherent in the source distribution. However, in real-world deployments, the model encounters a target stream where inputs $x$ are drawn from a shifted distribution $Q(x) \neq P(x)$. This distribution shift is often diverse, arising not only from specialized terminology in vertical domains (*e.g.*, medical or legal) but also from subtle variations in user intent and linguistic diversity in general contexts. Under such shifts, the model's learned dependency between context and subsequent tokens can degrade. Crucially, since LLMs generate sequences in an autoregressive manner, even a small distributional deviation early in a reasoning chain can propagate and amplify, pulling the model off a correct reasoning trajectory and producing coherent but factually incorrect outputs.

**Online Test-Time Learning (OTTL)** aims to adapt the model to the target distribution $Q(x)$ in real time, relying solely on the incoming test data stream $\mathcal{D}_{\text{stream}} = \{\mathcal{B}_1, \mathcal{B}_2, \ldots, \mathcal{B}_T\}$. Unlike the offline setting, OTTL processes data sequentially: at each time step $t$, the model

receives a batch $\mathcal{B}_t$, performs inference to generate predictions $\hat{y}$, and immediately updates its parameters before discarding the data. Formally, let $\mathcal{L}$ be an objective derived from the inputs, the LLM's own predictions, or a combination of both. The model parameters are then updated via gradient descent:

$$\Theta_t = \Theta_{t-1} - \eta \cdot \nabla_\Theta \mathcal{L}\left(f_{\Theta_{t-1}}(\mathcal{B}_t)\right), \qquad (1)$$

where $\eta$ is the learning rate. Directly minimizing the objective in Eq. (1) over all generated tokens assigns equal weight to correct reasoning steps and hallucinated content, which may reinforce errors. Thus, the core challenge is to identify and learn from only those tokens that contribute constructively to the reasoning process.

## 4. Future-Gain Guided Test-Time Learning

In this paper, we introduce **F**uture-**G**ain Guided **T**est-**T**ime Learning (**FG-TTL**), an online learning method that enhances target-domain performance by dynamically identifying and learning from only those informative tokens that contribute to reducing future uncertainty, using only unlabeled test data in a streaming setting. The overall workflow is illustrated in Figure 1, and the pseudo-code is in Algorithm 1. FG-TTL is composed of three key components: **1)** *Selective Token Entropy Minimization for OTTL.* Motivated by the distinct entropy dynamics of tokens during test time entropy minimization, we apply entropy minimization only to informative tokens, rather than applying uniform updates over the full sequence (*c.f.* Sec. 4.1). **2)** *Future-Gain Guided Token Selection.* To identify informative tokens, we introduce *Future-Gain*, a metric that quantifies the ex-

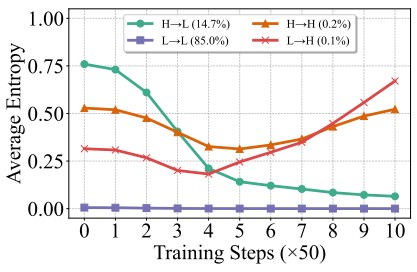

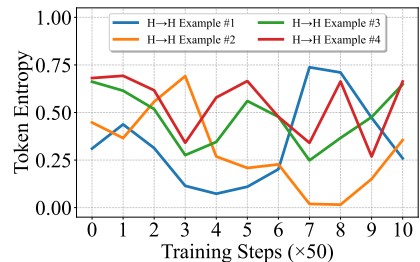

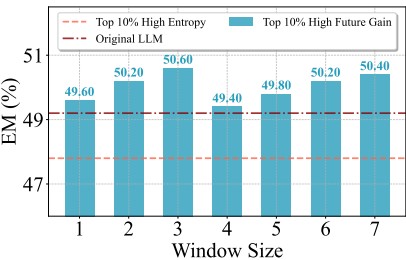

*(a)* Entropy dynamics of different tokens during test-time entropy minimization.

*(b)* Four cases of fluctuating tokens' entropy during test-time learning.

*(c)* Performance of different token selection strategies on MATH-500.

*Figure 2.* Summary of our exploration and observations: (a) reveals four distinct token entropy transition patterns during test-time learning using entropy minimization as the objective; (b) demonstrates that tokens falling into $H \rightarrow H$ category show high-volatility entropy throughout training, implying that the model has persistent uncertainty regarding these tokens; and (c) indicates that tokens that reduce future uncertainty contribute more to LLM performance improvement during test time.

pected reduction in future uncertainty contributed by each token. Inspired by human decision-making, which naturally prioritizes information that reduces future uncertainty, Future-Gain guides the selection of tokens that are more beneficial for improving future predictions. Based on this metric, we then develop a two-stage selection strategy that jointly considers Future-Gain and entropy, enabling updates to focus on tokens that are most beneficial for improving future predictions (*c.f.* Sec. 4.2). **3)** *Risk-Aware Adaptation.* We design a mechanism that combines gain-based weighting with adaptive temperature scaling to control the intensity of parameter updates. It ensures conservative adaptation on uncertain tokens while enabling aggressive learning from high-gain tokens (*c.f.* Sec. 4.3).

### 4.1. Selective Token Entropy Minimization for OTTL

Shannon entropy (Shannon, 1948) provides a fundamental measure of uncertainty in a probability distribution. In language modeling, it quantifies the model's predictive confidence: low entropy indicates high certainty about the next token, whereas high entropy reflects ambiguity or unfamiliarity with the context. Let $\mathbf{y} = (y_1, \ldots, y_L)$ denote a generated token sequence. At generation step $t$, the model $f_\Theta$ outputs a categorical distribution over the vocabulary $\mathcal{V}$, denoted $p_\Theta(y_t \mid \mathbf{y}_{<t})$. The token-level entropy of $\mathbf{y}$ is:

$$H(\mathbf{y}) = \frac{1}{L} \sum_{t=1}^{L} H_t, \qquad (2)$$

where the entropy at step $t$ is:

$$H_t := H\big(p_\Theta(\cdot \mid \mathbf{y}_{<t})\big) = -\sum_{v \in \mathcal{V}} p_\Theta(v \mid \mathbf{y}_{<t}) \log p_\Theta(v \mid \mathbf{y}_{<t}) \qquad (3)$$

Minimizing $H(\mathbf{y})$ encourages confident predictions on unlabeled test samples, which often improves target domain performance (Wang et al., 2021; Niu et al., 2022). However, directly applying entropy minimization (EM) to autoregressive LLM generation is non-trivial. Since EM relies on the

model's own predicted labels, it inherently encourages the model to amplify the logit of the most likely token, regardless of its correctness. Consequently, any noise or error in the generated sequence, which is inevitable, risks being reinforced through adaptation. This motivates a key insight: not all generated tokens are equally informative or worthy of parameter updates. To investigate this, we conduct a preliminary study and obtain the following observation:

**Observation 1: Tokens exhibit distinct entropy dynamics during test-time entropy minimization.** We analyze token-level entropy trajectories under full-sequence EM using an offline TTL setup (Hu et al., 2025) on MATH-500 (Lightman et al., 2024). Specifically, we apply EM to the entire output sequence of Llama3.1-8B-Instruct, save model checkpoints every 50 training steps, and compute the entropy of each response token at these intervals. From Figure 2a, tokens can be grouped into four categories based on their entropy evolution: 1) persistently high entropy ($H \rightarrow H$); 2) initially low but increasing entropy ($L \rightarrow H$); 3) initially high but steadily decreasing entropy ($H \rightarrow L$); and 4) consistently low entropy ($L \rightarrow L$). Quantitatively, only 14.7% of tokens belong to the $H \rightarrow L$ group, indicating expected uncertainty reduction. In contrast, the majority (85.0%) remain in the $L \rightarrow L$ category, suggesting they are already well-learned and contribute little new learning signal. Notably, 0.2% of the tokens are persistently high entropy ($H \rightarrow H$), likely because the model fails to resolve their ambiguity, while 0.1% of tokens show unexpected transitions from low to high entropy ($L \rightarrow H$) during training. Figure 2b illustrates that $H \rightarrow H$ tokens exhibit volatile, non-convergent entropy trajectories, implying that updating on them may reinforce unstable patterns. This challenges the practice of uniform entropy minimization, which treats redundant, erroneous and informative tokens identically.

Based on **Observation 1**, we design a selective token entropy minimization strategy that restricts updates to a reli-

able token subset $\mathcal{S}$:

$$\mathcal{L}(\Theta) = \frac{1}{|\mathcal{S}|} \sum_{t=1}^{L} \mathbb{I}(t \in \mathcal{S}) H_t \,, \quad (4)$$

where $\mathbb{I}(\cdot)$ is the indicator function, $\mathcal{S}$ is the set of positions corresponding to selected tokens, and $H_t$ is the entropy at position $t$. We optimize $\min_{\Theta} \mathcal{L}(\Theta)$ by back-propagating through LoRA, updating only $\Delta\Theta$ while keeping the base parameters frozen for stability (Hu et al., 2025).

## 4.2. Future-Gain Guided Token Selection

Inspired by human cognitive mechanisms where humans tend to prioritize actions that reduce future uncertainty (Bennett et al., 2016) and the information-theoretic notion of predictive information (Bialek et al., 2001), we hypothesize that informative tokens should similarly serve as informational pivots and reduce the uncertainty of subsequent token generation. To explore this, we conduct a preliminary study, leading to the following observation:

**Observation 2: Tokens that reduce future uncertainty contribute more to LLM performance improvement during test time.** Recent work (Wang et al., 2025b) updates the model only on high-entropy tokens to minimize immediate uncertainty. In contrast, we focus on tokens that enhance *future* predictive confidence (*i.e.* those that reduce the entropy of subsequent tokens). In autoregressive generation, each token is conditioned on the past and, once generated, reshapes the predictive distribution of the future. Thus, we heuristically define the Future-Gain of token $t$ as $\mathcal{G}_t = \frac{1}{W} \sum_{i=1}^{W} H_{t-i} - \frac{1}{W} \sum_{j=1}^{W} H_{t+j}$, where $[H_1, H_2, \ldots, H_L]$ is the sequence of predictive entropies and $W$ is a fixed window size. This metric measures the extent to which generating token $t$ reduces uncertainty about subsequent predictions. We compare two token selection strategies with the original LLM on the MATH-500 benchmark: 1) updating on the top-10% tokens with the highest entropy; 2) updating on the top-10% tokens with the highest *Future-Gain*, evaluated under varying window sizes to further investigate its impact; and 3) the original LLM without test-time updates. From Figure 2c, selecting tokens based on their ability to reduce future uncertainty and updating on them yields substantially higher accuracy (50.60% at $W = 3$), consistently outperforming both updating on high-entropy tokens and full-sequence updating across all window sizes, demonstrating that future-directed information flow provides a more reliable adaptation signal than immediate token confidence. We also find that performance is sensitive to the window size, which indicates that a fixed size cannot universally capture the information flow dynamics across different contexts.

Building on **Observation 2**, we propose a Future-Gain Guided Token Selection strategy to identify and prioritize

informational pivots for test-time updates by focusing on tokens that reduce downstream uncertainty. To this end, we introduce **Future-Gain**, which is derived from local entropy dynamics of the autoregressive chain, to capture the information flow within the generation trajectory. Specifically, let $H_t$ denote the predictive entropy at step $t$. For each position $t$, we first compute its local entropy volatility:

$$\sigma(H, t) = \sqrt{\frac{1}{2M} \sum_{k \in [-M, M], k \neq 0} \left( H_{t+k} - \bar{H}_{t,M} \right)^2}, \quad (5)$$

where $M$ is a fixed window size and $\bar{H}_{t,M} = \frac{1}{2M} \sum_{k \in [-M, M], k \neq 0} H_{t+k}$ is the average entropy over the $2M$ neighboring tokens. Then, we calculate an adaptive horizon $N(t)$ that scales inversely with local volatility:

$$N(t) = \min\left( N_{\max}, \; \max\left( N_{\min}, \; \left\lfloor \frac{\gamma}{\sigma(H, t) + \epsilon} \right\rfloor \right) \right), \quad (6)$$

where $N_{\min}$, $N_{\max}$, and $\gamma > 0$ are hyperparameters, and $\epsilon > 0$ ensures numerical stability. Finally, the Future-Gain at position $t$ is defined as:

$$\mathcal{G}_t = \underbrace{\frac{1}{N(t)} \sum_{i=1}^{N(t)} H_{t-i}}_{\text{Past Entropy (Prior Confusion)}} - \underbrace{\frac{1}{N(t)} \sum_{j=1}^{N(t)} H_{t+j}}_{\text{Future Entropy (Posterior Clarity)}}. \quad (7)$$

For boundary handling, we duplicate boundary entropy values when estimating local volatility, while excluding tokens without complete adaptive past and future windows from Future-Gain based updates. The two terms in Eq. (7) represent local prior confusion before position $t$ and realized downstream uncertainty after generating $y_t$, respectively. Their difference $\mathcal{G}_t$ therefore measures the empirical reduction in future uncertainty along the generated trajectory. Under a local stationarity assumption, $\mathcal{G}_t$ can also be interpreted as a plug-in surrogate for normalized conditional information gain; see Appendix A for the formal derivation and approximation bound. Moreover, the adaptive horizon endows FG with two key advantages: 1) it accounts for varying information density across different semantic regions; and 2) it captures longer-range information flow beyond the immediate next token.

**Selection Protocol.** Our token selection proceeds in two stages: (i) filter out trivial tokens with entropy below a threshold $\xi_{\text{low}}$; (ii) retain only the top-$\rho$ percentile of the remaining tokens ranked by $\mathcal{G}_t$. This protocol isolates informational pivots, namely tokens that exhibit meaningful uncertainty and significantly enhance future predictive clarity. By focusing adaptation exclusively on these high-impact tokens, we achieve more reliable and robust online learning.

## 4.3. Risk-Aware Adaptation

The Future-Gain guided selection strategy effectively identifies candidate informational pivots for learning, yet not all selected tokens are equally reliable. As shown in Figure 2b, some tokens exhibit persistent high-entropy oscillations, indicating that the model remains fundamentally uncertain about them. *Assigning a uniform weight to such unstable predictions may reinforce noise or erroneous patterns.*

To address this, we design a Risk-Aware Adaptation (RAA) mechanism that dynamically adjusts the strength of parameter updates based on the model's instantaneous uncertainty. *The core idea is simple*: For tokens that clearly reduce future uncertainty (high Future-Gain), we allow strong updates to leverage their high-value information; for tokens that remain highly uncertain, even if they have high Future-Gain, we suppress gradient flow to prevent model corruption.

We implement this principle through future-gain weighting and adaptive temperature scaling. First, each selected token $t \in \mathcal{S}$ is assigned a gain-based weight:

$$w_t = \frac{\exp(\mathcal{G}_t/\tau)}{\sum_{k \in \mathcal{S}} \exp(\mathcal{G}_k/\tau)}, \quad (8)$$

where $\tau > 0$ controls the sharpness of the weighting distribution. This ensures that tokens contributing more to downstream clarity dominate the learning signal. Second, to modulate update intensity, we define a temperature:

$$T_t = 1 + \alpha \cdot H(\mathbf{p}^{(t)}), \quad \alpha \geq 0, \quad (9)$$

where $H(\mathbf{p}^{(t)})$ is the entropy of the model's predictive distribution at step $t$. We then compute the temperature-scaled distribution $\mathbf{q}^{(t)}$ from the logits $\mathbf{z}^{(t)}$ via temperature-scaled softmax:

$$q_v^{(t)} = \frac{\exp(z_v^{(t)}/T_t)}{\sum_{k \in \mathcal{V}} \exp(z_k^{(t)}/T_t)}. \quad (10)$$

The final objective combines both components:

$$\mathcal{L}(\Theta) = \sum_{t=1}^{L} \mathbb{I}(t \in \mathcal{S}) \cdot w_t \cdot H(\mathbf{q}^{(t)}). \quad (11)$$

The theoretical soundness of this design is rooted in its implicit control over gradient magnitudes. As derived in Appendix B, we treat the selection mask $\mathbb{I}(t \in \mathcal{S})$, the gain-based weight $w_t$, and the temperature $T_t$ as stop-gradient control variables, which are computed from the forward pass and used only to modulate the adaptation loss. Considering the gradient of the loss with respect to the pre-softmax logits $\mathbf{z}^{(t)}$, the magnitude is scaled by the inverse temperature:

$$\|\nabla_{\mathbf{z}^{(t)}} \mathcal{L}\|_2 = \frac{1}{T_t} \cdot \|\mathbf{q}^{(t)} \odot (\log \mathbf{q}^{(t)} + H(\mathbf{q}^{(t)}))\|_2. \quad (12)$$

---

**Algorithm 1** The pipeline of proposed FG-TTL.

---

**Input:** Test samples $\mathcal{D}_{\text{stream}} = \{x_j\}_{j=1}^{M_{\text{test}}}$, the LLM $f_{\Theta_0}(\cdot)$, LoRA $\Delta\Theta$ with trainable parameters, batch size $B$.
1: Initialize LoRA parameters $\Delta\Theta$.
2: Add LoRA parameters to trained LLM $\tilde{\Theta} = \Theta_0 + \Delta\Theta$.
3: **for** a batch $\mathcal{B}_t = \{x_b\}_{b=1}^{B}$ in stream $\mathcal{D}_{\text{stream}}$ **do**
4:    Calculate predictions $\tilde{y}$ for all $x \in \mathcal{B}_t$ via $f_{\tilde{\Theta}}(\cdot)$.
5:    Calculate local volatility $\sigma(H, t)$ and dynamic window $N(t)$ via Eq. (6).
6:    Calculate Future-Gain $\mathcal{G}_t$ via Eq. (7).
7:    Select pivotal tokens based on gain ranking and entropy threshold.
8:    Calculate per-token weight $w_t$ and adaptive temperature $T_t$ via Eq. (8) and Eq. (9).
9:    Update LLM parameters $\Delta\Theta$ by minimizing Eq. (11).
10: **end for**
**Output:** The output answer $\{\hat{y}\}_{j=1}^{M_{\text{test}}}$ for all $x \in \mathcal{D}_{\text{stream}}$.

---

This formulation functions as a self-regulating control mechanism. Under confident generation (low $H(\mathbf{p}^{(t)})$), $T_t$ approaches 1, allowing the gain-based weight $w_t$ to drive effective parameter updates. Conversely, when the model is confused (high $H(\mathbf{p}^{(t)})$), $T_t$ increases proportionally, reducing the gradient magnitude through the factor $1/T_t$. This effectively gates the backpropagation, rendering the model inert to high-noise signals while preserving the capacity to learn from valid reasoning chains.

## 5. Experiments

### 5.1. Experimental Settings

**Datasets and Metrics.** We conduct experiments on six benchmarks that span a wide spectrum of domains and difficulty levels: GSM8K (Cobbe et al., 2021), MATH-500 (Lightman et al., 2024), CollegeMath (Tang et al., 2024), AIME24 (Zhang & Math-AI, 2024), Minerva (Lewkowycz et al., 2022), and OlympiadBench (He et al., 2024). These datasets cover grade school math, undergraduate science, and challenging competition problems. For all benchmarks, we report Exact Match (EM) accuracy (Chang et al., 2024).

**LLMs and Baselines.** We employ three Instruction-tuned LLMs with varying parameter scales: Qwen2.5-7B-Instruct (Yang et al., 2024), Llama3.1-8B-Instruct (Team, 2024) and Phi-4-14B (Abdin et al., 2024). We compare FG-TTL against TTA methods adapted for LLMs, including TENT (Wang et al., 2021), EATA (Niu et al., 2022), and COME (Zhang et al., 2025a), as well as the recent TTL method TLM (Hu et al., 2025). More details in Appendix E.

**Implementation Details.** We focus on the online setting with a batch size of 8. We employ the AdamW optimizer

*Table 1.* Comparison with methods on six benchmarks. Best performance is indicated in **bold**, and the second-best is underlined.

| Method | Pub. Year | GSM8K | MATH-500 | CollegeMath | AIME24 | Minerva | Olympiad | Average |
|---|---|---|---|---|---|---|---|---|
| Llama3.1-8B-Instruct | - | 82.18 | 49.20 | 25.00 | 3.33 | 20.96 | 14.29 | 32.49 |
| • TENT (Wang et al., 2021) | ICLR'21 | 82.64 | 49.20 | 24.42 | 10.00 | 22.06 | 14.07 | 33.73 |
| • EATA (Niu et al., 2022) | ICML'22 | 83.47 | 49.40 | 24.92 | 6.67 | 22.06 | 13.30 | 33.30 |
| • COME (Zhang et al., 2025a) | ICLR'25 | 83.17 | 48.80 | 25.42 | 10.00 | 21.53 | 13.41 | 33.72 |
| • TLM (Hu et al., 2025) | ICML'25 | **85.06** | 50.00 | 24.92 | 6.67 | 21.32 | 14.16 | 33.69 |
| • FG-TTL (Ours) | - | 83.40 | **51.60** | **25.92** | **13.33** | **22.79** | **14.95** | **35.33** |
| Qwen2.5-7B-Instruct | - | 82.56 | 72.40 | 22.75 | 10.00 | 27.94 | 31.21 | 41.14 |
| • TENT (Wang et al., 2021) | ICLR'21 | 83.24 | 73.40 | 23.00 | 10.00 | 28.68 | 32.08 | 41.73 |
| • EATA (Niu et al., 2022) | ICML'22 | 83.62 | 73.80 | 22.67 | 6.67 | 29.04 | 32.09 | 41.32 |
| • COME (Zhang et al., 2025a) | ICLR'25 | 81.50 | 73.40 | 22.67 | 6.67 | **29.41** | 32.09 | 40.96 |
| • TLM (Hu et al., 2025) | ICML'25 | 82.89 | 73.40 | 22.92 | 10.00 | 27.94 | 31.54 | 41.45 |
| • FG-TTL (Ours) | - | **84.31** | **74.40** | **23.17** | **13.33** | 29.04 | **32.31** | **42.76** |
| Phi-4-14B | - | 86.66 | 77.80 | 34.17 | 20.00 | 31.62 | 38.13 | 48.06 |
| • TENT (Wang et al., 2021) | ICLR'21 | 87.19 | **78.60** | 34.50 | 16.67 | 32.35 | 39.12 | 48.07 |
| • EATA (Niu et al., 2022) | ICML'22 | 87.41 | 78.00 | 34.50 | 16.67 | 32.72 | 39.01 | 48.05 |
| • COME (Zhang et al., 2025a) | ICLR'25 | 87.33 | 77.80 | 34.42 | 16.67 | 33.09 | 39.34 | 48.11 |
| • TLM (Hu et al., 2025) | ICML'25 | 87.26 | 76.40 | 34.17 | 16.67 | 32.72 | 39.23 | 47.74 |
| • FG-TTL (Ours) | - | **87.49** | **78.60** | **34.67** | **23.33** | **33.82** | **39.67** | **49.60** |

with LoRA applied to $W_q$ and $W_v$ modules. The learning rate is tuned for each dataset. For FG-TTL, we set the volatility sensitivity $\gamma = 1.0$, filter threshold $\xi_{low} = 1e-4$, and selection ratio $\rho = 20\%$. More details in Appendix F. The source code is available at https://github.com/BianLangyu/FG-TTL.git

## 5.2. Comparison Experiments

The main results on six reasoning benchmarks are summarized in Table 1. FG-TTL achieves consistent performance gains across all model scales, effectively outperforming both the original LLMs and state-of-the-art (SOTA) baselines.

**Significant Gains on Complex Reasoning Tasks.** As shown in Table 1, FG-TTL achieves competitive performance across most benchmarks under three representative LLM backbones. For Llama3.1-8B-Instruct, FG-TTL achieves the highest average score and delivers improvements on challenging reasoning tasks, including AIME24 ($10.00\% \rightarrow 13.33\%$) and Olympiad ($14.29\% \rightarrow 14.95\%$). For Qwen2.5-7B-Instruct, on GSM8K, our method exceeds the best competing method by $+0.69\%$ ($83.62\% \rightarrow 84.31\%$). On AIME24, FG-TTL outperforms the best baseline by $+3.33\%$ ($10.00\% \rightarrow 13.33\%$). This indicates that our Future-Gain Guided Token Selection successfully identifies and reinforces critical pivot tokens in the reasoning chain, thereby correcting the generation trajectory for complex problems where baselines often accumulate errors.

**Scalability Across Model Sizes.** As shown in Table 1, FG-TTL remains effective as model size increases and continues to provide consistent gains on the stronger Phi-4-14B backbone. Even when starting from a math-strong base

*Table 2.* Experimental results for the components of our proposed FG-TTL under Llama3.1-8B-Instruct.

| FTS | RAA | GSM8K | MATH-500 | CollegeMath | AIME24 | Average |
|---|---|---|---|---|---|---|
| | | 82.18 | 49.20 | 25.00 | 3.33 | 39.93 |
| ✓ | | 82.11 | 50.20 | 25.17 | 10.00 | 41.87 |
| | ✓ | 82.70 | 50.00 | 25.33 | 10.00 | 42.01 |
| ✓ | ✓ | **83.40** | **51.60** | **25.92** | **13.33** | **43.56** |

model with limited room for further improvement, FG-TTL achieves the best average EM. On the hardest benchmark AIME24, it is the only method that improves over the base model ($20.00\% \rightarrow 23.33\%$), while all competing baselines degrade performance to $16.67\%$. These results indicate that FG-TTL maintains consistent improvements as model capacity increases, and avoids the degradation observed in prior baselines on AIME24.

## 5.3. Ablation Studies

To verify the effectiveness of individual components within FG-TTL, we conduct ablation studies using Llama3.1-8B-Instruct. We present the analysis based on four representative datasets in this section, while the comprehensive results covering all six benchmarks are detailed in Appendix G.

**Contribution of Key Components.** We evaluate the incremental impact of our two primary components, the Future-Gain Guided Token Selection (FTS) strategy and Risk-Aware Adaptation (RAA) mechanism. As shown in Table 2, the original LLM achieves an average accuracy of $39.93\%$ on the evaluated subset. Adding **FTS** strategy alone improves the average accuracy to $41.87\%$, suggesting that focusing updates on informational pivots provides

*Table 3.* Comparison of selection strategies (Top-20%). *High Ent.* denotes selecting the most high-entropy tokens.

| Strategy | GSM8K | MATH-500 | CollegeMath | AIME24 | Average |
|---|---|---|---|---|---|
| LLM | 82.18 | 49.20 | 25.00 | 3.33 | 39.93 |
| Random | 80.44 | 49.33 | 25.39 | 10.00 | 41.29 |
| High Ent. | 80.06 | 50.40 | 24.83 | 6.67 | 40.49 |
| **Ours** | **83.40** | **51.60** | **25.92** | **13.33** | **43.56** |

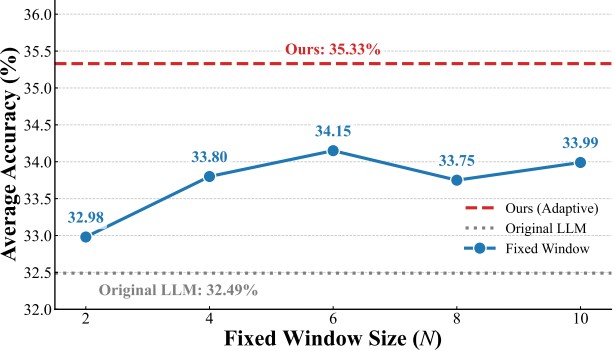

*Figure 3.* **Sensitivity analysis on window size.** The blue line indicates the performance of fixed window sizes. The red dashed line represents our adaptive window strategy. FG-TTL outperforms the best fixed configuration ($N = 6$) without manual tuning.

a higher-quality learning signal than adapting on the full sequence. Applying **RAA** mechanism alone also increases performance to $42.01\%$, indicating that uncertainty-aware modulation of update strength is important for stable adaptation. Combining both components yields the largest gain, reaching $43.56\%$, which confirms that deciding *where* to learn (FTS) and controlling *how strongly* to update (RAA) are complementary for robust online test-time learning.

**Effectiveness of Selection Strategy.** We investigate whether the improvement stems from sparse updating or from the Future-Gain criterion. We compare our strategy with Random Selection and High-Entropy Selection, while keeping the selection ratio fixed to the top-20% tokens. As shown in Table 3, selecting high-entropy tokens yields the smallest improvement, achieving an average accuracy of $40.49\%$ and underperforming random selection at $41.29\%$. This strongly implies that in unsupervised test-time settings, high uncertainty often correlates with noise or hallucinations rather than informative hard examples, and training on them reinforces errors. In contrast, FG-TTL achieves the best average accuracy of $43.56\%$, outperforming both alternatives and supporting our premise that effective update signals come from tokens that reduce downstream ambiguity rather than those that are merely unpredictable.

**Necessity of Adaptive Window.** We analyze the sensitivity of the window size $N(t)$ used in calculating Future-Gain (Figure 3). The performance with fixed window sizes exhibits a convex trend, peaking at $N = 6$. Windows that

*Table 4.* Experimental results for online test-time learning under the quantized setting using Llama3.1-8B-Instruct.

| Method | GSM8K | MATH-500 | CollegeMath | AIME24 | Average |
|---|---|---|---|---|---|
| LLM | 81.96 | 46.60 | 23.75 | 3.33 | 38.91 |
| TENT | 64.73 | 47.00 | 22.83 | 3.33 | 34.47 |
| EATA | 69.52 | 46.20 | 22.08 | 3.33 | 35.28 |
| COME | 68.23 | 45.00 | 23.33 | 3.33 | 34.97 |
| TLM | 82.49 | 46.20 | 24.08 | 3.33 | 39.03 |
| **Ours** | **83.32** | **47.60** | **24.42** | **6.67** | **40.50** |

*Table 5.* Comparison with SOTA methods on the DomainBench under Llama3.1-8B-Instruct. The metric is ROUGE-Lsum.

| Method | Geography | Agriculture | Medicine | Finance | Average |
|---|---|---|---|---|---|
| LLM | 24.41 | 8.76 | 13.56 | 22.51 | 17.31 |
| TENT | 25.36 | 6.24 | 14.48 | 21.40 | 16.87 |
| EATA | 25.14 | 6.26 | 13.82 | 18.86 | 16.02 |
| COME | **25.86** | 4.07 | 14.55 | 6.99 | 12.87 |
| TLM | 25.53 | **9.41** | 13.72 | 22.95 | 17.90 |
| **Ours** | 25.74 | 9.29 | **15.25** | **22.98** | **18.32** |

are too small fail to capture sufficient semantic dependencies, while excessively large windows introduce noise from unrelated future contexts. Crucially, our adaptive window strategy consistently outperforms the optimal fixed configuration. This confirms that the information density of reasoning chains is non-uniform, and our volatility-based mechanism effectively aligns the estimation horizon with the intrinsic granularity of the generation process, eliminating the need for manual hyperparameter tuning.

### 5.4. More Discussions

**Robustness under Quantization.** Following previous studies (Hu et al., 2025), we extend our evaluation to quantized LLMs to assess performance in resource-constrained scenarios. We conduct experiments using a 4-bit quantized version of Llama3.1-8B-Instruct, strictly adhering to the configurations outlined in QLoRA (Dettmers et al., 2023). As shown in Table 4, FG-TTL demonstrates robustness to quantization noise, achieving a SOTA average accuracy of $40.50\%$. This represents a substantial improvement over the $39.03\%$ achieved by the leading baseline TLM and the $38.91\%$ of the original LLM. These results confirm that the proposed FG-TTL remains effective and applicable even within the noisy parameter space of quantized models.

**Generalization to Vertical Domains.** Following previous studies (Hu et al., 2025), we evaluate domain knowledge learning on DomainBench. From Table 5, FG-TTL outperforms both the original LLM and TLM on average. Specifically, on the Medicine domain, FG-TTL achieves a substantial improvement by $1.69\%$ ($13.56\% \rightarrow 15.25\%$) over the base model. Furthermore, in terms of average performance,

*Table 6.* Efficiency and performance comparison on MATH-500 under Llama3.1-8B-Instruct. Latency is measured in seconds, extra memory in GB, throughput in tokens per second, and accuracy in %. Best results are in **bold**, and second-best results are underlined.

| Method | Latency ↓ | Memory ↓ | Throughput ↑ | Acc. ↑ |
|---|---|---|---|---|
| TENT | 40.42 | 14.31 | 31.58 | 49.20 |
| EATA | 38.43 | 13.98 | 31.97 | 49.40 |
| COME | **37.27** | 23.72 | 31.05 | 48.80 |
| TLM | 42.30 | **0.78** | 31.61 | 50.00 |
| **FG-TTL** | 39.30 | 13.61 | **32.01** | **51.60** |

our method yields a gain of 1.01% (17.31%→18.32%), whereas some baselines such as EATA and COME reduce the performance. These results indicate that FG-TTL effectively leverages domain-specific knowledge while maintaining stability, particularly in specialized fields like Medicine and Finance.

**Efficiency Analysis.** To evaluate the practical deployment cost of FG-TTL, we further compare its time and memory efficiency with all test-time adaptation and test-time learning baselines. We conduct the efficiency analysis on MATH-500 using Llama3.1-8B-Instruct under the same online setting as the main experiments. We report four metrics: total latency, extra GPU memory consumption, decoding throughput, and final accuracy. Total latency measures the end-to-end wall-clock time for online inference and adaptation. Extra memory denotes the additional peak GPU memory compared with the frozen inference model. Throughput is measured as the number of generated tokens per second. As shown in Table 6, FG-TTL introduces only moderate wall-clock overhead. Its total latency is 39.30s, which is comparable to EATA (38.43s) and lower than TLM (42.30s). Although FG-TTL performs token selection and risk-aware adaptation during generation, the additional computation does not lead to substantial latency degradation.

## 6. Conclusion

In this paper, we propose **FG-TTL**, an online learning method that enhances target-domain performance using only unlabeled test data. Motivated by cognitive principles of uncertainty reduction and the information-theoretic notion of predictive information, we focus on tokens that act as informational pivots and clarify the downstream decoding trajectory. Specifically, we develop a **FTS** strategy to decide where to learn by prioritizing tokens that reduce downstream uncertainty in subsequent generations. We introduce *Future-Gain* as a token-level metric for this purpose and update the model only on high-gain tokens, concentrating learning on informative positions and mitigating temporal error propagation. In addition, we design a **RAA** mechanism that controls how strongly to update. It modulates

update strength through gain-based weighting and adaptive temperature scaling based on intrinsic uncertainty, which reduces gradient magnitude when the model is unreliable and enables stronger learning on high-gain tokens.

## Acknowledgments

This work was partially supported by the Joint Funds of the National Natural Science Foundation of China under Grant No. U24A20327, and the Guangdong S&T Program under Grant No. 2026B0101110001.

## Impact Statement

This paper presents work whose goal is to advance the field of Machine Learning. There are many potential societal consequences of our work, none of which we feel must be specifically highlighted here.

## Author Contributions

Langyu Bian and Jinwu Hu contributed equally to this work. Qing Du, Qi Chen, and Mingkui Tan are the corresponding authors. Langyu Bian and Jinwu Hu conceived the main idea and designed the proposed method. Langyu Bian conducted the experiments and performed the main empirical analysis. Langyu Bian, Jinwu Hu, Zitian Zhang, Dongjin Yang, and Yufeng Wang contributed to manuscript writing, result organization, and paper revision. Qing Du, Qi Chen, and Mingkui Tan supervised the project, provided guidance on methodology and experiments, and revised the manuscript.

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

# Supplementary Materials for
## "Future-Gain Guided Test-Time Learning for Large Language Models"

## A. Theoretical Interpretation of Future-Gain

In this section, we provide an information-theoretic interpretation of Future-Gain and show that it serves as a computationally tractable surrogate for the conditional information gain that a generated token provides about the future trajectory.

### A.1. Setup

Consider an autoregressive language model parameterized by $\Theta$. At generation position $t$, let

$$C_t = (x, Y_{<t}) \tag{13}$$

denote the historical context, where $x$ is the input prompt and $Y_{<t}$ are the previously generated tokens. Let $Y_t$ be the current token and define the future window of length $N$ as

$$F_t^{(N)} = (Y_{t+1}, Y_{t+2}, \ldots, Y_{t+N}). \tag{14}$$

The amount of information that $Y_t$ carries about the future trajectory, conditioned on the past context, can be measured by the conditional mutual information:

$$I_\Theta(Y_t; F_t^{(N)} \mid C_t) = H_\Theta(F_t^{(N)} \mid C_t) - H_\Theta(F_t^{(N)} \mid C_t, Y_t). \tag{15}$$

Intuitively, the first term measures the uncertainty of the future before observing the current token, while the second term measures the remaining future uncertainty after observing the current token. Therefore, a token with large conditional mutual information substantially reduces future uncertainty and is a desirable target for test-time learning.

For a realized generated trajectory, we define the token-level predictive entropy as

$$H_s = -\sum_{v \in \mathcal{V}} p_\Theta(v \mid x, y_{<s}) \log p_\Theta(v \mid x, y_{<s}), \tag{16}$$

where $\mathcal{V}$ is the vocabulary and $y_{<s}$ is the realized prefix.

### A.2. From Conditional Information Gain to Future-Gain

Directly computing Eq. (15) is infeasible during online test-time learning, because evaluating $H_\Theta(F_t^{(N)} \mid C_t)$ requires marginalizing over all possible current tokens and future continuations. We therefore approximate it using entropy statistics along the observed generation trajectory.

First, by the chain rule of entropy, the posterior future uncertainty after observing the current token can be decomposed as

$$H_\Theta(F_t^{(N)} \mid C_t, Y_t = y_t) = \sum_{j=1}^{N} H_\Theta\left(Y_{t+j} \mid C_t, y_t, Y_{t+1:t+j-1}\right). \tag{17}$$

Along a realized generated trajectory, the average of the future token entropies provides a plug-in estimate of the normalized posterior uncertainty:

$$\frac{1}{N} H_\Theta(F_t^{(N)} \mid C_t, Y_t = y_t) \approx \frac{1}{N} \sum_{j=1}^{N} H_{t+j}. \tag{18}$$

The prior future uncertainty $H_\Theta(F_t^{(N)} \mid C_t)$ is more difficult to compute, since it corresponds to the uncertainty of future trajectories before fixing the current token. We approximate this term using a local stationarity assumption: within a local reasoning segment, the model's intrinsic uncertainty varies smoothly unless an informative pivot token appears. Under this assumption, the average entropy of the immediate past window serves as a tractable proxy for the normalized prior future uncertainty:

$$\frac{1}{N} H_\Theta(F_t^{(N)} \mid C_t) \approx \frac{1}{N} \sum_{i=1}^{N} H_{t-i}. \tag{19}$$

Substituting Eq. (18) and Eq. (19) into the normalized conditional information gain gives

$$\frac{1}{N} I_\Theta(Y_t; F_t^{(N)} \mid C_t) \approx \frac{1}{N} \sum_{i=1}^{N} H_{t-i} - \frac{1}{N} \sum_{j=1}^{N} H_{t+j}$$

$$= \mathcal{G}_t. \tag{20}$$

This is exactly the Future-Gain score used in our method when the horizon is fixed to $N$. In the main method, we further replace the fixed horizon $N$ with the adaptive horizon $N(t)$, which allows the estimator to account for varying information propagation ranges across different reasoning regions.

## A.3. Formal Approximation Bound

We next make the above approximation explicit.

**Assumption A.1** (Local stationarity and plug-in estimation). For each position $t$ and horizon $N$, assume that the following approximation errors are bounded:

$$\left| \frac{1}{N} H_\Theta(F_t^{(N)} \mid C_t) - \frac{1}{N} \sum_{i=1}^{N} H_{t-i} \right| \le \epsilon_{\text{past}}, \tag{21}$$

$$\left| \frac{1}{N} H_\Theta(F_t^{(N)} \mid C_t, Y_t = y_t) - \frac{1}{N} \sum_{j=1}^{N} H_{t+j} \right| \le \epsilon_{\text{future}}. \tag{22}$$

Here, $\epsilon_{\text{past}}$ captures the local stationarity error of using past entropies as a prior proxy, and $\epsilon_{\text{future}}$ captures the plug-in estimation error of using the realized future trajectory.

**Theorem A.2** (Future-Gain as a CMI surrogate). *Under Assumption A.1, the Future-Gain score*

$$\mathcal{G}_t = \frac{1}{N} \sum_{i=1}^{N} H_{t-i} - \frac{1}{N} \sum_{j=1}^{N} H_{t+j} \tag{23}$$

*approximates the normalized conditional information gain of the current token about the future trajectory:*

$$\left| \mathcal{G}_t - \frac{1}{N} \left[ H_\Theta(F_t^{(N)} \mid C_t) - H_\Theta(F_t^{(N)} \mid C_t, Y_t = y_t) \right] \right| \le \epsilon_{\text{past}} + \epsilon_{\text{future}}. \tag{24}$$

*Moreover, taking expectation over $Y_t \sim p_\Theta(\cdot \mid C_t)$ recovers the normalized conditional mutual information:*

$$\mathbb{E}_{Y_t \mid C_t} \left[ H_\Theta(F_t^{(N)} \mid C_t) - H_\Theta(F_t^{(N)} \mid C_t, Y_t) \right] = I_\Theta(Y_t; F_t^{(N)} \mid C_t). \tag{25}$$

*Thus, Future-Gain is a computationally feasible surrogate for selecting tokens with high conditional information about future generation.*

*Proof.* By definition,

$$\mathcal{G}_t = \frac{1}{N} \sum_{i=1}^{N} H_{t-i} - \frac{1}{N} \sum_{j=1}^{N} H_{t+j}. \tag{26}$$

Adding and subtracting the normalized prior and posterior future entropies yields

$$\mathcal{G}_t - \frac{1}{N}\left[H_\Theta(F_t^{(N)} \mid C_t) - H_\Theta(F_t^{(N)} \mid C_t, Y_t = y_t)\right] = A_t - B_t, \tag{27}$$

where

$$A_t = \frac{1}{N}\sum_{i=1}^{N} H_{t-i} - \frac{1}{N}H_\Theta(F_t^{(N)} \mid C_t), \tag{28}$$

$$B_t = \frac{1}{N}\sum_{j=1}^{N} H_{t+j} - \frac{1}{N}H_\Theta(F_t^{(N)} \mid C_t, Y_t = y_t). \tag{29}$$

By the triangle inequality and Assumption A.1,

$$\left| \mathcal{G}_t - \frac{1}{N}\left[H_\Theta(F_t^{(N)} \mid C_t) - H_\Theta(F_t^{(N)} \mid C_t, Y_t = y_t)\right] \right| \le |A_t| + |B_t| \le \epsilon_{\text{past}} + \epsilon_{\text{future}}. \tag{30}$$

Finally, by the definition of conditional mutual information,

$$I_\Theta(Y_t; F_t^{(N)} \mid C_t) = H_\Theta(F_t^{(N)} \mid C_t) - H_\Theta(F_t^{(N)} \mid C_t, Y_t), \tag{31}$$

where the second term is understood as an expectation over $Y_t \sim p_\Theta(\cdot \mid C_t)$. This completes the proof. $\qquad\square$

**Discussion.** The above result shows that Future-Gain is not an ad-hoc heuristic. Instead, it approximates the reduction in future uncertainty induced by the current token. Tokens with large $\mathcal{G}_t$ are therefore likely to be informational pivots that clarify subsequent generation. This provides an information-theoretic justification for using Future-Gain to guide token selection during test-time learning.

## B. Theoretical Analysis of Risk-Aware Gradient Scaling

In this section, we provide a rigorous theoretical foundation for our Risk-Aware Adaptation mechanism by analyzing how adaptive temperature scaling modulates gradient magnitudes during test-time learning.

**Notations.** Let $\mathbf{z} \in \mathbb{R}^{|V|}$ denote the pre-softmax logits produced by the language model, where $|V|$ is the vocabulary size. The risk-aware adaptation mechanism applies temperature scaling with parameter $T_t \ge 1$ to these logits before computing the softmax distribution $\mathbf{q} \in \Delta^{|V|-1}$. The objective function is the Shannon entropy $\mathcal{L} = H(\mathbf{q}) = -\sum_{i=1}^{|V|} q_i \log q_i$.

**Remark.** The partial derivative of $\mathcal{L}$ with respect to model parameter is $\nabla_\theta \mathcal{L}$, which can be decomposed as $\nabla_\theta \mathcal{L} = (\partial \mathbf{z}/\partial \theta)^T \cdot \nabla_\mathbf{z} \mathcal{L}$. Since $\partial \mathbf{z}/\partial \theta$ is independent of $T_t$, analyzing $\nabla_\mathbf{z} \mathcal{L}$ isolates the specific effect of temperature scaling on gradient magnitudes. To this end, our analysis focuses on characterizing the relationship between the temperature parameter $T_t$ and the gradient magnitude $\|\nabla_\mathbf{z} \mathcal{L}\|_2$.

### B.1. Preliminaries and Assumptions

*Assumption* 1. (**Differentiability of softmax**) The temperature-scaled softmax function $q_i = \frac{\exp(z_i/T_t)}{\sum_j \exp(z_j/T_t)}$ is continuously differentiable with respect to $\mathbf{z}$ for all $T_t > 0$ and $\mathbf{z} \in \mathbb{R}^{|V|}$.

Assumption 1 ensures the existence of the Jacobian matrix for the softmax transformation, which is fundamental to our gradient analysis. This assumption holds universally for temperature-scaled softmax functions as long as $T_t > 0$.

*Assumption* 2. (**Non-degenerate probability distribution**) The probability distribution $\mathbf{q}$ induced by temperature scaling satisfies $\min_i q_i > 0$, ensuring the entropy $H(\mathbf{q})$ is well-defined and differentiable.

Assumption 2 guarantees numerical stability in entropy computation and is naturally satisfied for finite logits and positive temperature parameters.

## B.2. Derivation of Gradient Scaling

We now present the rigorous derivation of the gradient norm with respect to the pre-softmax logits.

*Theorem* 1. (**Temperature-dependent gradient scaling**) Under Assumptions 1 and 2, the L2-norm of the gradient of entropy loss with respect to pre-softmax logits is given by:

$$\|\nabla_{\mathbf{z}}\mathcal{L}\|_2 = \frac{1}{T_t} \|\mathbf{q} \odot (\log \mathbf{q} + H(\mathbf{q}))\|_2, \tag{32}$$

where $\odot$ denotes element-wise multiplication, $\mathbf{q}$ is the temperature-scaled probability distribution, and $H(\mathbf{q})$ is its entropy.

*Proof.* Following the chain rule, the partial derivative of $\mathcal{L}$ with respect to logit $z_k$ is:

$$\frac{\partial \mathcal{L}}{\partial z_k} = \sum_{i=1}^{|V|} \frac{\partial \mathcal{L}}{\partial q_i} \frac{\partial q_i}{\partial z_k}. \tag{33}$$

We compute each component separately. First, the derivative of entropy with respect to probability:

$$\frac{\partial \mathcal{L}}{\partial q_i} = \frac{\partial}{\partial q_i} \left( -\sum_{j=1}^{|V|} q_j \log q_j \right) = -(\log q_i + 1). \tag{34}$$

Second, the Jacobian of the temperature-scaled softmax function has elements:

$$\frac{\partial q_i}{\partial z_k} = \frac{1}{T_t} q_i(\delta_{ik} - q_k), \tag{35}$$

where $\delta_{ik}$ is the Kronecker delta function ($\delta_{ik} = 1$ if $i = k$, otherwise 0). This follows directly from differentiating the softmax function with temperature parameter $T_t$.

Substituting Equations (34) and (35) into Equation (33):

$$\begin{aligned}
\frac{\partial \mathcal{L}}{\partial z_k} &= \sum_{i=1}^{|V|} -(\log q_i + 1) \cdot \frac{1}{T_t} q_i(\delta_{ik} - q_k) \\
&= -\frac{1}{T_t} \left[ q_k(\log q_k + 1) - \sum_{i=1}^{|V|} q_i q_k(\log q_i + 1) \right] \\
&= -\frac{1}{T_t} \left[ q_k(\log q_k + 1) - q_k \sum_{i=1}^{|V|} q_i \log q_i - q_k \sum_{i=1}^{|V|} q_i \right] \\
&= -\frac{1}{T_t} \left[ q_k(\log q_k + 1) + q_k H(\mathbf{q}) - q_k \right] \\
&= -\frac{1}{T_t} q_k(\log q_k + H(\mathbf{q})).
\end{aligned}$$

Therefore, the gradient vector is:

$$\nabla_{\mathbf{z}}\mathcal{L} = -\frac{1}{T_t} \left[ q_1(\log q_1 + H(\mathbf{q})), \ldots, q_{|V|}(\log q_{|V|} + H(\mathbf{q})) \right]^\top. \tag{36}$$

Taking the L2-norm of both sides of Equation (36) yields:

$$\|\nabla_{\mathbf{z}}\mathcal{L}\|_2 = \frac{1}{T_t} \|\mathbf{q} \odot (\log \mathbf{q} + H(\mathbf{q}))\|_2, \tag{37}$$

which completes the proof. □

### B.3. Theoretical Implications

Theorem 1 indicates fundamental properties of the risk-aware adaptation mechanism:

*Corollary* 1. (**Gradient attenuation under high uncertainty**) As $T_t \to \infty$ (corresponding to high model uncertainty), the gradient norm satisfies:

$$\lim_{T_t \to \infty} \|\nabla_{\mathbf{z}} \mathcal{L}\|_2 = 0.$$

*Proof.* As $T_t \to \infty$, the probability distribution $\mathbf{q}$ approaches uniformity with $q_i \to \frac{1}{|V|}$ for all $i$. Consequently:

$$\log q_i + H(\mathbf{q}) \to \log\left(\frac{1}{|V|}\right) + \log|V| = 0.$$

Since both terms in $\|\mathbf{q} \odot (\log \mathbf{q} + H(\mathbf{q}))\|_2$ remain bounded while $\frac{1}{T_t} \to 0$, the gradient norm converges to zero. $\square$

Corollary 1 establishes that our risk-aware mechanism automatically suppresses parameter updates in high-noise regions (high uncertainty) while preserving learning capacity in reliable regimes. This self-regulating property is crucial for maintaining model stability when processing out-of-distribution inputs or ambiguous reasoning steps.

The theoretical analysis demonstrates that adaptive temperature scaling creates an implicit control mechanism that modulates learning intensity based on intrinsic model uncertainty, which enables more stable and robust online learning.

## C. More Related Work

**Test-Time Adaptation (TTA)** adjusts a trained model during inference using unlabeled test data to better match the test distribution and maintain reliable performance under distribution shift (Wang et al., 2021; Niu et al., 2022; 2023; Wang et al., 2025d; Liang et al., 2025; Zhang et al., 2025a; Li et al., 2025). TENT (Wang et al., 2021) introduces a test-time adaptation strategy that reduces prediction uncertainty by minimizing entropy during inference while updating only normalization-related components. Following the need for efficiency and robustness, EATA (Niu et al., 2022) develops an economical adaptation pipeline that selects reliable low-entropy samples for updates and employs a Fisher-based regularizer with pseudo-labels to protect important parameters from catastrophic forgetting, while SAR (Niu et al., 2023) complements this by filtering out noisy, high-gradient test examples and applying sharpness-aware optimization to steer the model toward flatter, more robust minima.

Addressing temporal and continual shifts, CoTTA (Wang et al., 2022) enables long-term adaptation in non-stationary target streams via weight- and augmentation-averaged predictions together with stochastic neuron restoration to retain source knowledge. NOTE (Gong et al., 2022) strengthens robustness to temporally correlated non-i.i.d. streams by correcting normalization per instance and constructing class-balanced i.i.d.-like buffers through prediction-balanced reservoir sampling. MEMO (Zhang et al., 2022) pursues test-time robustification by minimizing the entropy of the marginal prediction across multiple augmentations of each input, enforcing invariance and consistent confident predictions without relying on data streams. RoTTA (Yuan et al., 2023) further stabilizes adaptation in dynamic, gradually changing environments with robust normalization statistics, a time- and uncertainty-aware memory bank for balanced sampling, and a teacher–student time-aware reweighting scheme. Finally, CoLA (Chen et al., 2024a) complements these single-device methods by establishing a cross-device collaborative paradigm that accumulates shared domain-knowledge vectors and enables either reprogramming-based adaptation on resource-rich agents or similarity-driven aggregation on lightweight followers to boost adaptation efficiency and accuracy across devices.

## D. More Details for Datasets

To comprehensively evaluate the robustness and effectiveness of FG-TTL, we conduct experiments on six benchmarks that represent a diverse spectrum of difficulty levels, knowledge domains, and temporal distributions. These datasets are publicly available and we strictly adhere to their respective usage licenses.

**GSM8K** (Cobbe et al., 2021). As a fundamental baseline for multi-step mathematical reasoning, GSM8K consists of high-quality grade school math word problems created by human solvers. We utilize the standard test split containing 1,319 samples. The problems require 2 to 8 reasoning steps involving basic arithmetic operations, testing the model's ability to

maintain logical consistency over short trajectories. The dataset is released under the *MIT License* and can be accessed at `https://github.com/openai/grade-school-math`.

**MATH-500** (Lightman et al., 2024). This dataset is a curated evaluation subset derived from the larger MATH benchmark, widely used to benchmark strong reasoning models. It comprises 500 unique problems sampled from the test set, covering seven distinct mathematical disciplines: Prealgebra, Algebra, Number Theory, Counting & Probability, Geometry, Intermediate Algebra, and Precalculus. The difficulty aligns with high school competitions. We access the data via Hugging Face under the *MIT License* at `https://huggingface.co/datasets/HuggingFaceH4/MATH-500`.

**CollegeMath** (Tang et al., 2024). To evaluate the model's proficiency in undergraduate-level mathematics, we utilize the CollegeMath benchmark, which covers seven core disciplines including calculus, linear algebra, and probability theory. For our experimental setup, we randomly sampled 1,200 instances from the original test set. This sampling strategy ensures an efficient yet representative evaluation of the model's advanced mathematical reasoning capabilities without incurring excessive computational overhead. The dataset is released under the Creative Commons Attribution 3.0 Unported License (CC BY 3.0) and can be accessed at `https://huggingface.co/datasets/di-zhang-fdu/College_Math_Test`.

**AIME24** (Zhang & Math-AI, 2024). This dataset represents the upper echelon of high school mathematical competition, containing 30 highly complex problems from the 2024 American Invitational Mathematics Examination. Unlike standard tasks, these problems require extended Chain-of-Thought (CoT) reasoning and deep logical coherency. We utilize this benchmark specifically to stress-test the model's stability over long generation trajectories, assessing whether the adaptation method can maintain correct reasoning paths without drifting. The data is accessible at `https://huggingface.co/datasets/HuggingFaceH4/aime_2024`.

**Minerva** (Lewkowycz et al., 2022). This benchmark is designed to evaluate interdisciplinary scientific reasoning beyond pure mathematics. containing 272 undergraduate-level science and mathematics problems sourced from MIT OpenCourseWare, the dataset rigorously tests the model's capability to handle specialized scientific terminology and perform complex quantitative problem-solving in fields such as physics, chemistry, and electrical engineering. The dataset is released under an unspecified license but is publicly available for non-commercial research purposes, adhering to fair use guidelines from MIT OpenCourseWare sources, and can be accessed at `https://huggingface.co/datasets/math-ai/minervamath`.

**OlympiadBench** (He et al., 2024). As a comprehensive bilingual benchmark sourced from top-tier international competitions, OlympiadBench challenges the limits of logical reasoning. To focus strictly on language-based reasoning adaptation, we filter the dataset to exclude multimodal inputs. Specifically, we aggregate the `OE_TO_maths_en_COMP` subset (674 samples) and the `OE_TO_physics_en_COMP` subset (236 samples), resulting in a total of 910 text-only instances. This selection allows us to evaluate the method's performance on open-ended, high-difficulty problems. The processed data can be obtained from `https://huggingface.co/datasets/Hothan/OlympiadBench`.

# E. More Details for Experiment Settings

Given that several baseline Test-Time Adaptation (TTA) methods originated in discriminative computer vision tasks, we adapt their distinct optimization objectives to the autoregressive generation paradigm of Large Language Models (LLMs). To guarantee a rigorous and fair comparison, all methods utilize Low-Rank Adaptation (LoRA) for parameter updates, replacing method-specific parameters.

**TENT** (Wang et al., 2021) operates on the principle of entropy minimization to enhance model confidence on unlabeled test data. While originally designed to update batch normalization statistics in convolutional networks, we transpose this objective to the sequence generation domain. Specifically, we minimize the mean Shannon entropy of the predictive distribution averaged over all generated tokens in the sequence. This approach serves as a fundamental baseline for self-supervised adaptation, enforcing high confidence in the model's own generation trajectory.

**EATA** (Niu et al., 2022) builds upon entropy minimization by introducing an active sample selection mechanism and Fisher regularization to mitigate noisy gradient updates. The core premise is that high-entropy samples are likely to induce degradation rather than facilitate adaptation. In our LLM implementation, we incorporate a sequence-level filtration strategy. We calculate the average entropy of the generated response; if this metric exceeds a pre-determined threshold $E_0 = 0.4$, the sample is excluded from the backward optimization pass. This mechanism effectively prevents the model from reinforcing

*Table 7.* Full Component Ablation. Comparison of FG-TTL variants on all six benchmarks. "w/o RAA" denotes removing Adaptive Temperature Scaling (using fixed temperature), and "w/o FTS" denotes removing Future-Gain Selection (training on all tokens or using standard selection).

| Method | GSM8K | MATH-500 | CollegeMath | AIME24 | Minerva | Olympiad | Average |
|---|---|---|---|---|---|---|---|
| Llama3.1-8B-Instruct | 82.18 | 49.20 | 25.00 | 3.33 | 20.96 | 14.29 | 32.49 |
| w/o RAA (FTS only) | 82.11 | 50.20 | 25.17 | 10.00 | 21.69 | 13.74 | 33.82 |
| w/o FTS (RAA only) | 82.70 | 50.00 | 25.33 | 10.00 | 22.43 | 13.63 | 34.02 |
| **FG-TTL (Ours)** | **83.40** | **51.60** | **25.92** | **13.33** | **22.79** | **14.95** | **35.33** |

*Table 8.* Full Strategy Comparison. Comparison of different token selection strategies (top-20% ratio) on all six benchmarks. "High Entropy" selects tokens with the highest uncertainty, while "Random" selects tokens uniformly at random.

| Selection Strategy | GSM8K | MATH-500 | CollegeMath | AIME24 | Minerva | Olympiad | Average |
|---|---|---|---|---|---|---|---|
| Llama3.1-8B-Instruct | 82.18 | 49.20 | 25.00 | 3.33 | 20.96 | 14.29 | 32.49 |
| Random Selection (20%) | 80.44 | 49.33 | 25.39 | 10.00 | 21.57 | 14.11 | 33.47 |
| High Entropy (20%) | 80.06 | 50.40 | 24.83 | 6.67 | 21.69 | 14.50 | 33.03 |
| **FG-TTL (Ours)** | **83.40** | **51.60** | **25.92** | **13.33** | **22.79** | **14.95** | **35.33** |

low-confidence hallucinations.

**COME** (Zhang et al., 2025a) addresses the overconfidence and model collapse issues inherent in standard entropy minimization by introducing a Conservative Entropy objective. This method explicitly models uncertainty via a Dirichlet prior distribution over the predictions. A critical modification for the LLM context involves the definition of the category space. We scale the conservative regularization term to accommodate the extensive vocabulary size of LLMs, as opposed to the limited class count in standard classification tasks. This adaptation penalizes the distribution for collapsing into single-token repetitions, thereby maintaining generation diversity.

**TLM** (Hu et al., 2025) represents a Test-Time Learning paradigm tailored specifically for LLMs. Distinct from standard TTA approaches that optimize based on pseudo-labels derived from model outputs, TLM focuses on aligning the model with the linguistic distribution of the target domain. We implement this by formulating the objective as input perplexity minimization. The model parameters are updated to maximize the likelihood of the input prompt tokens using the standard causal language modeling loss. This strategy supports domain alignment without relying on potentially erroneous generated answers. Furthermore, we adopt the Sample Efficient Learning Strategy proposed in the original work to prioritize high-perplexity prompts that offer significant informational gain.

## F. Hyperparameter Settings

**General Training Configuration.** All experiments are conducted in an online setting where test samples arrive sequentially. We use a streaming batch size of 8 and the AdamW optimizer. For parameter-efficient fine-tuning, we apply LoRA with a rank $r = 8$ and $\alpha = 16$. We apply LoRA to the $W_q$ and $W_v$ modules for all models, with the exception of Phi-4-14B, where we target the $W_q$, $W_k$, and $W_v$ modules. Matrix $\mathcal{A}$ is initialized with random Gaussian values, while matrix $\mathcal{B}$ is set to zero. To ensure deterministic evaluation, we employ greedy decoding with a temperature of 0 across all experiments. The infrastructure consists of NVIDIA A800 GPUs (80GB) running PyTorch 2.7.0.

**Dataset-Specific Learning Rates.** To ensure optimal training stability, the learning rates were tuned specifically for each dataset across different models. For **Llama3.1-8B-Instruct**, the learning rate is set to $1.0 \times 10^{-8}$ for GSM8K and CollegeMath, $7.5 \times 10^{-6}$ for MATH-500, AIME24, and Minerva, and $5.0 \times 10^{-6}$ for OlympiadBench. Regarding **Qwen2.5-7B-Instruct**, we utilized $1.0 \times 10^{-5}$ for GSM8K, $7.5 \times 10^{-6}$ for MATH-500 and Minerva, $1.0 \times 10^{-6}$ for OlympiadBench, and a lower rate of $1.0 \times 10^{-7}$ for CollegeMath and AIME24. Finally, for **Phi-4-14B**, the rates are configured as $1.5 \times 10^{-5}$ for CollegeMath, $1.0 \times 10^{-5}$ for GSM8K, $7.5 \times 10^{-6}$ for AIME24 and Minerva, and $5.0 \times 10^{-6}$ for both MATH-500 and OlympiadBench.

**FG-TTL Specific Parameters.** For our proposed method, we set the volatility sensitivity parameter $\gamma = 1.0$, fixed window size $M = 6$ and bound the adaptive window size $N(t)$ within the range $[2, 10]$. We filter out trivial tokens with entropy

*Table 9.* Detailed results of window size sensitivity analysis on Llama3.1-8B-Instruct. We compare our adaptive window strategy with various fixed window sizes ($N \in \{2, 4, 6, 8, 10\}$). The best performance in each column is highlighted in **bold**.

| Method | GSM8K | MATH-500 | CollegeMath | AIME24 | Minerva | Olympiad | Average |
|---|---|---|---|---|---|---|---|
| Original Model | 82.18 | 49.20 | 25.00 | 3.33 | 20.96 | 14.29 | 32.49 |
| Fixed Window ($N = 2$) | 82.34 | 50.00 | 24.75 | 6.67 | 21.69 | 12.42 | 32.98 |
| Fixed Window ($N = 4$) | 83.09 | 49.80 | 25.33 | 10.00 | 21.69 | 12.86 | 33.80 |
| Fixed Window ($N = 6$) | 82.18 | 51.20 | 25.00 | 10.00 | **22.79** | 13.74 | 34.15 |
| Fixed Window ($N = 8$) | 82.94 | 50.20 | 24.23 | 10.00 | 20.96 | 14.18 | 33.75 |
| Fixed Window ($N = 10$) | 83.09 | 50.40 | 25.42 | 10.00 | 20.96 | 14.07 | 33.99 |
| **FG-TTL (Adaptive)** | **83.40** | **51.60** | **25.92** | **13.33** | **22.79** | **14.95** | **35.33** |

below $\xi_{\text{low}} = 1e-4$ and retain the top $20\%$ of the remaining tokens ranked by Future-Gain. The temperature scaling factor $\alpha$ in the risk-aware update mechanism is set to $0.5$.

## G. More Details for Ablation Studies

**Component effectiveness across the full benchmark.** The ablation study across all benchmarks indicates distinct interaction patterns between our two core modules. On moderately difficult tasks such as GSM8K and MATH-500, both FTS alone and RAA alone yield comparable improvements over the base model, indicating that each component provides independent value under mild distribution shifts. However, on the most challenging benchmark AIME24, the synergy between components becomes critical. While each ablated variant achieves $10.00\%$ accuracy, their combination yields $13.33\%$, representing a $33.3\%$ relative improvement over either component in isolation. This demonstrates that for complex multi-step reasoning where error propagation risks are highest, precise token selection must be coupled with uncertainty-aware update control to stabilize the adaptation process. Notably, on Olympiad where base performance is already low, RAA alone slightly underperforms the base model at $13.63\%$, confirming that gradient modulation without informative token selection cannot compensate for fundamentally unstable generation trajectories.

**Selection strategy robustness across diverse reasoning tasks.** Future-Gain selection maintains consistent superiority across all benchmarks, with its largest margin observed on AIME24 where it outperforms random selection by $3.33$ percentage points. Critically, high-entropy selection underperforms random selection on AIME24, achieving only $6.67\%$ versus $10.00\%$. This failure mode confirms that tokens with high instantaneous entropy often reflect persistent confusion rather than informative reasoning pivots. The consistent advantage of Future-Gain selection across benchmarks with varying difficulty levels validates our core hypothesis: tokens that resolve downstream ambiguity constitute higher-quality supervision signals than those merely exhibiting high uncertainty at the current step.

**Task-dependent window sensitivity and adaptive horizon advantage.** Fixed window sizes exhibit strong task dependency in optimal horizon length. Window size $N = 6$ achieves peak performance on Minerva at $22.79\%$, while $N = 10$ performs best on CollegeMath at $25.42\%$. Crucially, no single fixed window dominates across all benchmarks; for instance, $N = 2$ yields $82.34\%$ on GSM8K but degrades to $12.42\%$ on Olympiad. This heterogeneity validates our adaptive horizon design. The volatility-based mechanism automatically adjusts $N(t)$ per token position, achieving $83.40\%$ on GSM8K and $14.95\%$ on Olympiad simultaneously. Most significantly, on AIME24 where all fixed windows plateau at $10.00\%$, our adaptive strategy reaches $13.33\%$, demonstrating that complex reasoning chains benefit substantially from context-aware horizon estimation that aligns with local information density variations.

## H. Additional Experimental Results

In this section, we provide additional experimental results to further evaluate the robustness, decoding stability, generalization to open-ended generation, hyperparameter sensitivity, and forgetting behavior of FG-TTL.

**Robustness under Noisy Online Streams.** To evaluate whether FG-TTL remains robust when the online stream contains corrupted inputs, we conduct a noisy mixed-stream stress test on MATH-500. Specifically, we randomly corrupt $20\%$ of the test samples by severely perturbing $30\%$ of their words, including random deletion, word swapping, and replacement

*Table 10.* Robustness under a noisy mixed-stream stress test on MATH-500.

| Method | Acc. |
|---|---|
| Base | 44.80 |
| TENT | 43.00 |
| EATA | 42.80 |
| COME | 44.00 |
| TLM | 42.20 |
| FG-TTL (Ours) | **45.60** |

*Table 11.* Impact of the token selection ratio $\rho$.

| Dataset | 10% | 20% | 30% | 50% |
|---|---|---|---|---|
| GSM8K | 83.02 | **83.40** | 82.79 | 82.94 |
| MATH-500 | 49.80 | **51.60** | 50.80 | 51.20 |

with random ASCII strings. This setting simulates hostile or low-quality online traffic where unreliable inputs may induce harmful test-time updates. As shown in Table 10, FG-TTL achieves the best accuracy under the noisy mixed-stream setting. Compared with entropy-minimization baselines, FG-TTL is less affected by corrupted inputs, suggesting that Future-Gain guided token selection and risk-aware adaptation help suppress harmful updates from noisy or unreliable samples.

**Sensitivity to the Token Selection Ratio.** We study the sensitivity of FG-TTL to the token selection ratio $\rho$, which controls the proportion of candidate tokens retained after Future-Gain ranking. A larger $\rho$ allows more tokens to contribute to the update, while a smaller $\rho$ makes the update more selective. Table 11 shows that the best performance is obtained at $\rho = 20\%$, while nearby choices remain competitive. This indicates that FG-TTL is not overly sensitive to the exact token selection ratio. In practice, selecting a moderate subset of high-gain tokens provides a good balance between extracting useful adaptation signals and avoiding noisy updates from less informative tokens.

**Sensitivity to the Entropy Threshold.** We evaluate the sensitivity of FG-TTL to the entropy threshold $\xi_{\text{low}}$. This threshold is used as a validity filter to remove near-deterministic tokens before Future-Gain ranking. The main token selection effect is still determined by Future-Gain rather than by entropy thresholding. As shown in Table 12, FG-TTL remains stable across different values of $\xi_{\text{low}}$. Although $\xi_{\text{low}} = 10^{-4}$ gives the best performance, the overall variation is moderate. This supports our design that the entropy threshold mainly filters trivial low-entropy tokens, while Future-Gain ranking provides the primary criterion for identifying informative update positions.

**Sensitivity to the Temperature Scaling Factor.** We analyze the sensitivity of FG-TTL to the temperature scaling factor $\alpha$ in the Risk-Aware Adaptation mechanism. Recall that $\alpha$ controls how strongly predictive entropy modulates the adaptive temperature $T_t = 1 + \alpha \cdot H(\mathbf{p}^{(t)})$. A smaller $\alpha$ weakens uncertainty-based gradient suppression, while a larger $\alpha$ makes the update more conservative on uncertain tokens. As shown in Table 13, FG-TTL remains stable across different values of $\alpha$, with the best performance achieved at $\alpha = 0.5$. When $\alpha$ is too small, the adaptive temperature provides weaker suppression for uncertain predictions, allowing noisier gradients to contribute to the update. When $\alpha$ is too large, the update becomes overly conservative, which may weaken adaptation on informative high-gain tokens. Overall, these results indicate that $\alpha = 0.5$ provides a balanced trade-off between adaptation strength and update stability.

**Results under Non-Greedy Decoding.** The main experiments use greedy decoding for deterministic and fair comparison. To examine whether FG-TTL remains effective under stochastic decoding, we conduct additional experiments with temperature 0.6 and top-$p = 0.9$ on MATH-500 and Minerva. Table 14 shows that FG-TTL consistently achieves the best performance under non-greedy decoding. This suggests that the proposed Future-Gain guided token selection and risk-aware update mechanism are not restricted to deterministic decoding, and remain effective when the generation process becomes more stochastic.

**Generalization to Open-Ended Generation.** We further evaluate FG-TTL on WritingPrompts to examine whether the proposed future-aware adaptation principle remains useful beyond constrained reasoning tasks. We randomly sample 100 examples and report ROUGE-L, BERTScore, and their arithmetic mean. Unlike mathematical reasoning tasks, open-ended generation may admit multiple plausible continuations rather than a single deterministic reasoning trajectory. This setting therefore tests whether Future-Gain can still identify useful update positions when the generation space is less constrained.

*Table 12.* Impact of the entropy threshold $\xi_{\text{low}}$.

| Dataset | $10^{-3}$ | $10^{-4}$ | $10^{-5}$ |
|---|---|---|---|
| GSM8K | 83.09 | **83.40** | 82.94 |
| MATH-500 | 50.20 | **51.60** | 50.00 |

*Table 13.* Impact of the temperature scaling factor $\alpha$.

| Dataset | 0.1 | 0.2 | 0.5 | 0.7 | 1.0 |
|---|---|---|---|---|---|
| GSM8K | 82.26 | 83.02 | **83.40** | 82.49 | 82.79 |
| MATH-500 | 50.20 | 50.00 | **51.60** | 49.80 | 51.00 |

As shown in Table 15, FG-TTL achieves the highest scores among the compared methods, although the gains are modest. These results suggest that FG-TTL is not restricted to tasks with a unique answer, and that Future-Gain guided adaptation can also provide useful update signals for open-ended generation.

**Analysis of Catastrophic Forgetting.** We examine whether test-time adaptation causes catastrophic forgetting of general knowledge. We first evaluate the original LLM on Agriculture to obtain the reference performance. Then, we adapt the model separately on each mathematical reasoning benchmark and evaluate the adapted model again on Agriculture. As shown in Table 16, the performance on Agriculture remains highly stable after adaptation on different mathematical reasoning datasets. The average score is unchanged, and all variations are small. These results indicate that FG-TTL can adapt to incoming reasoning streams through lightweight LoRA updates without substantially overwriting the model's existing knowledge.

# I. Qualitative Analysis of Token Selection

We analyze the tokens selected by FG-TTL for parameter updates on GSM8K and MATH-500. By comparing their linguistic properties and vocabulary composition against high-entropy (Top-20%) and low-entropy (Bottom-20%) baselines, we observe a distinct shift from surface-level formatting to semantic reasoning content.

## I.1. Semantic Preference Analysis

We classify the selected tokens into six categories: Syntax/LaTeX, Function words, Numbers, Verbs, Nouns, and Adjectives. As shown in Figure 4, the distinct semantic preferences of different strategies.

The Low-Entropy baseline mainly selects Syntax/LaTeX (37.83%) and Numbers (25.89%). This indicates that confidence-based selection favors formatting symbols and simple numeric completions. These tokens provide limited helpful information for reasoning. Similarly, the High-Entropy baseline is heavily distracted by Syntax/LaTeX (33.04%) and Function words (28.81%). High uncertainty on these tokens often reflects aleatoric uncertainty arising from linguistic variability rather than genuine logical ambiguity (Kendall & Gal, 2017). In contrast, FG-TTL allocates a significantly larger proportion to semantic tokens. Aggregating Nouns, Verbs, and Adjectives, our method selects 37.75% of tokens in this group, compared to only 19.85% for the Low-Entropy baseline. Specifically, Nouns account for 28.73% and Verbs for 5.33%. These tokens typically correspond to domain entities and critical reasoning steps in the chain-of-thought process. Updating on these tokens provides more informative signals for the subsequent decoding trajectory (Tishby et al., 2000).

## I.2. Token Composition Analysis

As shown in Figure 5, the overlap between tokens selected by FG-TTL and the entropy-based baselines. The composition indicates two distinct characteristics of our selection strategy.

**Rejection of Trivial Tokens.** Only 2.3% of the selected tokens overlap with the low-entropy set. This confirms that FG-TTL effectively filters out trivial and highly predictable tokens. These tokens offer little benefit for adaptation.

**Discovery of Hidden Pivots.** Approximately 14.5% of the semantic tokens are unique to our method. We term these tokens *hidden pivots*. Although they may not exhibit extreme instantaneous entropy, they are critical for reducing future predictive

*Table 14.* Results under non-greedy decoding with temperature 0.6 and top-$p = 0.9$.

| Method | MATH-500 | Minerva |
|---|---|---|
| Base | 49.80 | 22.79 |
| TENT | 47.20 | 22.79 |
| EATA | 48.40 | 22.06 |
| COME | 49.40 | 20.56 |
| TLM | 49.60 | 21.69 |
| FG-TTL (Ours) | **50.20** | **24.63** |

*Table 15.* Results on open-ended generation using WritingPrompts.

| Method | ROUGE-L | BERTScore | Avg. |
|---|---|---|---|
| Base | 0.1370 | 0.4954 | 0.3162 |
| TENT | 0.1357 | 0.4943 | 0.3150 |
| EATA | 0.1360 | 0.4947 | 0.3154 |
| COME | 0.1354 | 0.4930 | 0.3142 |
| TLM | 0.1363 | 0.4946 | 0.3155 |
| FG-TTL (Ours) | **0.1375** | **0.4960** | **0.3168** |

uncertainty. The remaining 45.3% consists of common terms selected by all methods.

### I.3. Vocabulary Analysis

The word clouds in Figure 6 visualize the vocabulary differences across strategies. The tokens unique to FG-TTL frequently include logical connectors and structural markers. These tokens typically precede a reduction in predictive entropy. They act as information bottlenecks that resolve ambiguity for downstream generation (Tishby et al., 2000). Conversely, the baselines are dominated by frequent function words and formatting symbols. These tokens are less likely to influence the reasoning path. This qualitative evidence supports the hypothesis that Future-Gain prioritization captures the semantic backbone of the reasoning chain rather than surface-level completions.

## J. Discussions and Future Work

In this paper, we propose a selective adaptation method for autoregressive generation that prioritizes tokens expected to reduce downstream uncertainty. Empirical results on six diverse benchmarks show consistent improvements over existing test-time learning methods, especially on complex multi-step reasoning tasks where error propagation is a major challenge.

**Deployment Considerations.** Beyond empirical effectiveness, practical deployment of online test-time learning requires careful consideration of serving constraints. FG-TTL is primarily designed for scenarios where online adaptability is important and moderate adaptation overhead is acceptable, such as on-device personalized agents, autonomous agents operating in changing environments, and offline enterprise batch-processing pipelines. In these settings, lightweight LoRA updates provide a practical mechanism for adapting to recurring distribution shifts without requiring labels or external retrieval corpora.

For high-throughput serving systems, FG-TTL can be implemented in an asynchronous shadow-updating manner. The latency-critical inference engine can remain forward-only to preserve KV-cache reuse and continuous batching, while a background adaptation process computes gradients on selected high-gain tokens and periodically synchronizes lightweight LoRA parameters back to the serving engine. This design separates adaptation from user-facing generation and makes FG-TTL more compatible with serving systems where online learning should not block inference. Nevertheless, integrating online parameter updates into highly optimized production LLM serving stacks remains an important systems challenge for future work.

**Future Directions.** A promising direction is to extend the Future-Gain principle to multimodal large language models (MLLMs). MLLMs such as LLaVA (Liu et al., 2023) generate text tokens autoregressively, and therefore, the core idea of identifying tokens that clarify later predictions can transfer naturally. Extending to multimodal systems requires careful calibration of entropy measures across modalities. Nevertheless, the central insight remains: effective adaptation should

*Table 16.* Performance on Agriculture before and after TTL on mathematical reasoning datasets.

|  | GSM8K | MATH-500 | College | AIME24 | Minerva | Olympiad | Avg. |
|---|---|---|---|---|---|---|---|
| Before | 8.76 | 8.76 | 8.76 | 8.76 | 8.76 | 8.76 | 8.76 |
| After | 8.71 | 8.85 | 8.79 | 8.75 | 8.87 | 8.56 | 8.76 |
| Δ | -0.05 | +0.09 | +0.03 | -0.01 | +0.11 | -0.20 | 0.00 |

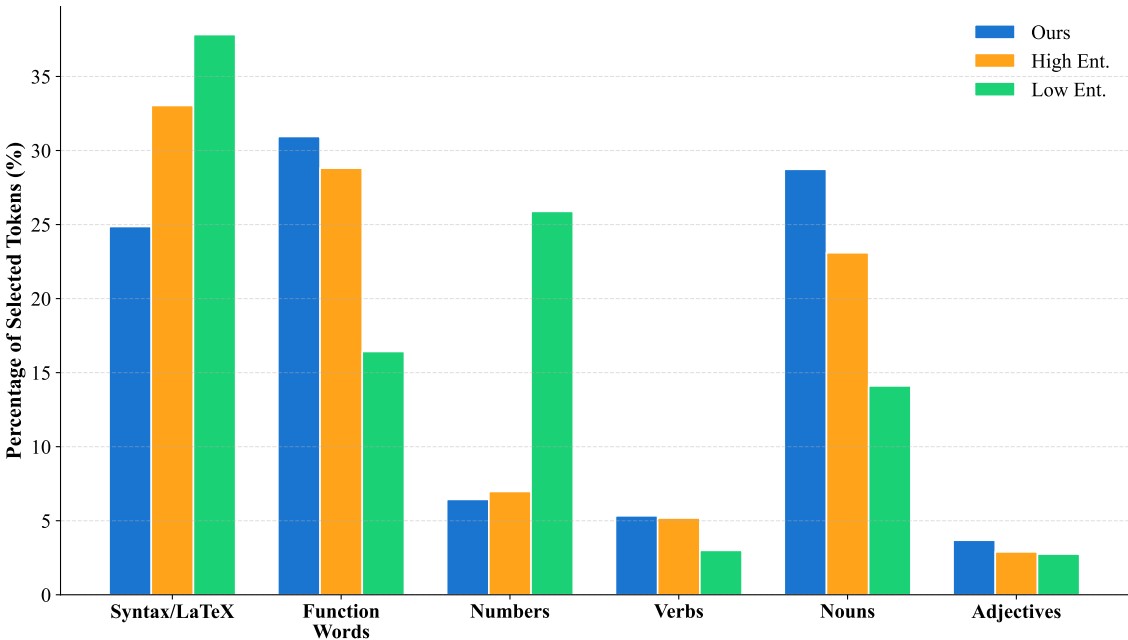

*Figure 4.* Token type distribution under different selection strategies. We categorize the selected tokens by their semantic roles and report the percentage in each category. Compared with entropy-based baselines that concentrate on syntax-related tokens or trivial numbers, FG-TTL selects a more balanced set of reasoning-relevant content tokens (*e.g.*, nouns, verbs, and adjectives), indicating a stronger focus on the semantic core of the problem.

target information flow rather than instantaneous confidence.

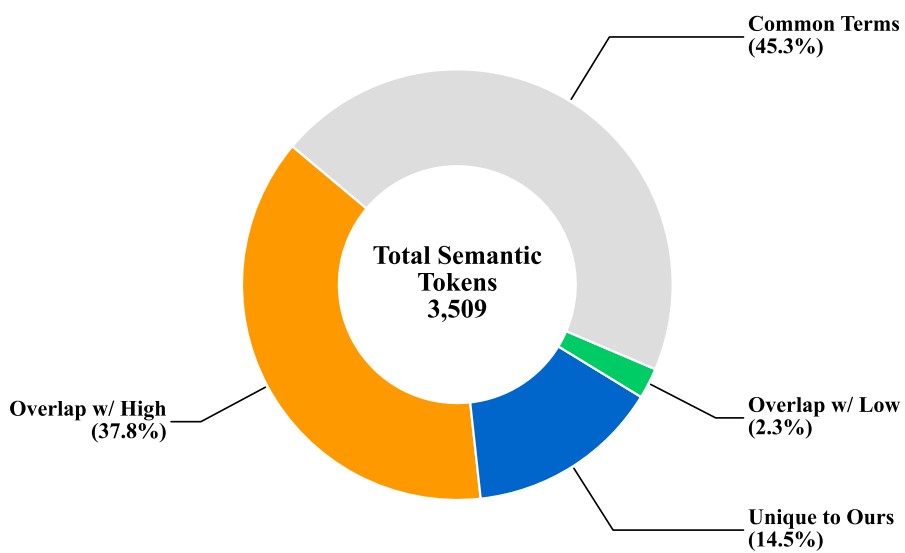

*Figure 5.* Composition of Selected Tokens. Our method retains hard examples (overlap with High Ent.) representing epistemic uncertainty, while discovering unique "Hidden Pivots" (14.5%) that reduce future uncertainty. Trivial tokens (overlap with Low Ent.) are largely rejected.

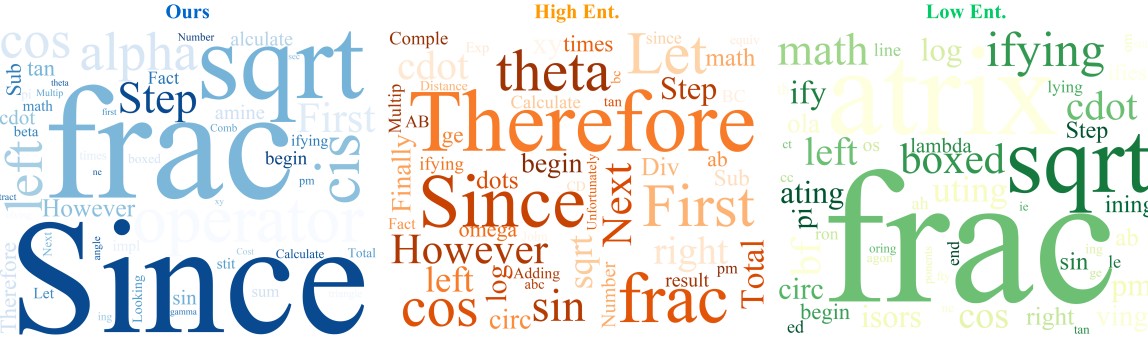

*Figure 6.* Vocabulary Analysis via Word Clouds. The visualization confirms that FG-TTL (Left) prioritizes mathematical concepts and logical connectors, whereas High-Entropy (Center) and Low-Entropy (Right) baselines are cluttered with general function words and LaTeX symbols.

