# OpenReview forum: "Future-Gain Guided Test-Time Learning for Large Language Models"
_ICML.cc/2026/Conference — ICML 2026 regular_

### Official Review · Reviewer_3YB7 · 2026-03-02

**Soundness:** 3
**Presentation:** 3
**Significance:** 2
**Originality:** 2
**Overall Recommendation:** 4
**Confidence:** 4

**Summary:**

The paper proposes Future-Gain Guided Test-Time Learning (FG-TTL), a test-time learning framework for LLMs that adapts LoRA parameters online using only unlabeled test streams. Instead of minimizing entropy on all generated tokens, FG-TTL computes a token-level "Future-Gain" that measures how much a token reduces future predictive entropy, selects high-gain tokens, and performs selective entropy minimization. A Risk-Aware Adaptation mechanism further modulates update strength via gain-based weights and entropy-driven temperature scaling, suppressing gradients on uncertain tokens.

**Compliance With Llm Reviewing Policy:**

Affirmed.

**Final Justification:**

This paper addresses an important problem and is clearly presented, with generally solid empirical results. The rebuttal was helpful and clarified several technical points, but it did not fully resolve my main concerns. I remain unconvinced about the strength of the conceptual novelty, as the method still appears closer to a careful combination of selective entropy-based updating and uncertainty-aware gradient modulation than to a fundamentally new adaptation principle. I also still find the theoretical grounding of Future-Gain insufficiently principled, with Eq. (7) looking more like a reasonable empirical surrogate than a well-justified measure of token informativeness. Overall, while I appreciate the authors’ effort and the paper’s practical strengths, my final assessment remains cautious and only slightly changed after the rebuttal.

**Key Questions For Authors:**

1. How exactly are the local volatility (Eq. 5) and Future-Gain metric (Eq. 7) computed near sequence boundaries where the required token windows fall out of bounds, and does your chosen approach—whether padding, truncation, or token exclusion—inadvertently omit or bias the adaptation of critical early reasoning steps?

2. Regarding the Risk-Aware Adaptation, does the temperature-scaled loss in Eq. 11 fully replace the standard entropy objective from Eq. 4 during backpropagation, and specifically, is the dynamic temperature $T_t$ detached from the computational graph or do gradients actively flow through it?

3. Given the use of fixed selection hyperparameters—such as an extremely low entropy threshold of 1e-4 that seemingly filters very few tokens and a static 20\% selection ratio. How sensitive is the overall performance to these values, and were competing baselines afforded the exact same rigorous, dataset-specific tuning to ensure a perfectly fair comparison?

4. Conceptually, how does FG-TTL's token-selection strategy compare with recent structural regularization methods (such as the asymmetric network approach in ZeroSiam) for preventing test-time entropy collapse? Could structural alignment serve as a complementary mechanism to Future-Gain, potentially reducing the reliance on heuristic entropy filtering thresholds?

**Limitations:**

yes

**Strengths And Weaknesses:**

## Strengths

1. Measuring a token's usefulness by its ability to reduce future entropy is a meaningful conceptual contribution for autoregressive test-time adaptation. Eq. (7) defines Future-Gain as the difference between averaged past and future entropies under an adaptive horizon $N(t)$, effectively encoding an "informational pivot" that proxies for trajectory-steering tokens.
2. The Future-Gain metric (Eq. 7) and adaptive horizon (Eq. 6) naturally align with autoregressive information flow better than standard per-token entropy. Furthermore, the Risk-Aware Adaptation (Eqs. 8–11) effectively modulates learning intensity via a gain-based softmax and entropy-based temperature scaling, successfully learning strongly from reliable pivots and weakly from noisy ones.

## Weaknesses

1. Conceptual novelty relative to existing LLM test-time adaptation/TTL work is under-positioned and somewhat incremental. The paper clearly builds on entropy-minimization-based TTA (Tent, EATA, COME) and LLM TTL (TLM, SyTTA, TTRL), but the core change is: (i) a more carefully designed selection criterion (Future-Gain via local entropy before/after a token with adaptive window), and (ii) per-token temperature scaling based on entropy. This is an interesting combination, but not a fundamentally new paradigm: selection based on stability or future impact, and temperature/uncertainty-based gradient modulation, have precedents in the TTA and robust training literature. The related work section and Appendix B focus heavily on CV TTA, while more recent LLM-specific test-time scaling or verifier-guided improvement methods with selective updates (e.g., using verifier scores, intrinsic confidence, or RL-guided test-time updates) are not discussed, which makes the claimed conceptual space look wider than it is.

2. The formulation of the entropy-based loss is partially inconsistent. Section 4.1 frames the objective as minimizing the standard entropy $H_t$ of the original distribution $p_\Theta$. However, Section 4.3 and Eq. (11) define the loss using the entropy of a temperature-scaled distribution $q^{(t)}$. The text argues high entropy of $p^{(t)}$ increases temperature $T_t$ to attenuate gradients, but lacks explanation on why the entropy of $q$ is preferred over $p$ in the loss. Additionally, Eq. (10) has a likely typo in the denominator ($\sum_{k\in|\mathcal{V}|}$ instead of $v\in \mathcal{V}$), and the dependence of $H(\mathbf{p}^{(t)})$ on $\Theta$ is ignored during gradient discussions.


3. Selection protocol and thresholds feel ad hoc and under-motivated. The selection procedure first filters tokens with entropy below $\xi_{\text{low}}$, then keeps top $\rho$ percentile by $G_t$. However, $\xi_{\text{low}} = 1\mathrm{e}^{-4}$ (Appendix E) is extremely small, so in practice almost all tokens will pass this filter because token-level entropies for LLM vocabularies are rarely that low. That means the entropy threshold hardly plays the conceptual role of excluding trivial tokens, contrary to the narrative. Likewise, the choice of $\rho = 20\%$ is fixed across datasets; while Tab. 3 / 7 show that Future-Gain beats other selection rules given 20\% sparsity, there is no sensitivity analysis on $\rho$. Given that token sparsity directly affects computational efficiency and stability, some exploration of $\rho$ and $\xi_{\text{low}}$ is necessary to validate robustness and clarify whether the gains are sensitive to this hyperparameter.

4. Theorem 1 and Corollary 1 rigorously handle gradient scaling via temperature, but there is no theoretical analysis or at least principled justification of why the Future-Gain metric should correlate with "informativeness" or performance improvement beyond the empirical plots in Fig. 2(c). For example, one could have related $G_t$ to mutual information between $y_t$ and $y_{>t}$ under some simplifying assumptions, or at minimum, argued why this particular symmetric windowed difference is preferable to causal quantities like $H(y_{>t} \mid y_{\le t}) - H(y_{>t} \mid y_{<t})$.

5. The paper uses greedy decoding with temperature 0 across the board (Appendix E). Since the method is heavily driven by entropy patterns, it is not obvious how behavior changes under more realistic sampling (top-p, temperature $> 0$), especially since RL-based or verifier-based test-time methods often operate in those regimes. All adaptation updates are applied via LoRA on attention projections only, potentially benefiting methods reliant on fine-grained token-level entropy manipulations.

6. Several key hyperparameters are fixed across all datasets and models (e.g., $\gamma=1.0$, $M=6$, $\xi_{\text{low}}=$ 1e-4, $\rho=$ 20\%, $N_{\min}=2$, $N_{\max}=10$). Conversely, dataset-specific learning rates are heavily tuned per dataset and model. It is unclear if baselines received equally careful per-dataset tuning or used fixed learning rates from prior work, which could bias the comparisons.

---

> ### Author Rebuttal · Authors · 2026-03-31
>
> We are deeply grateful to you for recognizing the strengths of our work, particularly the " **meaningful conceptual contribution**", and "**better than standard per-token entropy**".
>
> >Q1. Novelty over prior TTL/TTA is unclear.
>
> **A1.** FG-TTL extends entropy-based TTL with a future-aware and reliability-aware adaptation principle for autoregressive generation:
> * **Future-aware adaptation.** Future-Gain selects tokens by downstream uncertainty reduction rather than current uncertainty alone.
> * **Reliable updates.** RAA uses gain-based weighting and entropy-aware temperature scaling to suppress harmful gradients on uncertain tokens.
> * **Relation to prior work.** We agree that the related work can better cover recent LLM-specific methods, and we will revise this. Unlike recent methods that rely on external verifiers, rewards, or auxiliary models, our setting is strictly source-free and uses only intrinsic signals from the unlabeled test stream.
>
> >Q2. Entropy loss is partially inconsistent and needs clarification.
>
> **A2.** We clarify the roles of p and q.
> * **Conceptual objective vs. implementation.** Sec. 4.1 presents the standard EM based on distribution p, while Sec. 4.3 introduces our risk-aware implementation using the temperature-scaled distribution q. Here, p estimates instantaneous uncertainty, and q defines the final controlled update objective.
> * **Practical Gradient Control through Temperature Scaling.** We use the entropy of q instead of p because temperature scaling directly controls gradient magnitude. Under high uncertainty, a larger temperature flattens q and suppresses noisy gradients.
> * **Notation correction.** Eq. 10 contains a typo: the denominator should sum over vocabulary elements $v \in \mathcal{V}$.
>
> >Q3. Sensitivity to $\rho$ and $\xi_{low}$ is unclear.
>
> **A3.** The performance of FG-TTL is not overly sensitive to $\rho$ and $\xi_{low}$.
> * **Sensitivity of $\rho$.** The best result is at $\rho=20%$, but nearby choices remain competitive.
> * **Sensitivity of $\xi_{low}$.** The best result  at $\xi_{low}=10^{-4}$, while overall performance remains stable. In practice, it mainly filters trivial low-entropy tokens, and the main selection effect comes from $G_t$.
>
> Tab. A: Impact of $\rho$
> |$\rho$|10%|20%|30%|50%|
> |-|-|-|-|-|
> |GSM8K|83.02|**83.40**|82.79|82.94|
> |MATH500|49.80|**51.60**|50.80|51.20|
>
> Tab. B: Impact of $\xi_{low}$
> |$\xi_{low}$|1e-3|1e-4|1e-5|
> |-|-|-|-|
> |GSM8K|83.09|**83.40**|82.94|
> |MATH500|50.20|**51.60**|50.00|
>
>
> >Q4. Theory of Future-Gain is insufficient.
>
> **A4.** Future-Gain is a theoretically motivated and tractable surrogate for informative update signals.
> * **Connection to conditional mutual information.** The causal quantity corresponds to conditional mutual information, implying that a token is more informative if it reduces future uncertainty.
> * **Tractable approximation.** Since the exact causal quantity is intractable online, Future-Gain approximates it along the realized decoding path.
>
> >Q5. Robustness beyond greedy decoding remains unclear.
>
> **A5.** We use greedy decoding for deterministic and fair comparison, and additional results on non-greedy decoding show that FG-TTL remains the best-performing method.
>
> Tab C: Results under Non-Greedy Decoding (Temp=0.6, Top-p=0.9)
> |Method|MATH500|Minerva|
> |-|-|-|
> |Base|49.80|22.79|
> |Tent|47.20|22.79|
> |EATA|48.40|22.06|
> |COME|49.40|20.56|
> |TLM|49.60|21.69|
> |Ours|**50.20**|**24.63**|
>
> >Q6. The fairness of baseline tuning is unclear.
>
> **A6.** For all baselines, we evaluate multiple learning rates, including the same learning-rate candidates used for FG-TTL, and report their best performance to ensure a fair comparison.
>
> >Q7. Boundary handling for local volatility and Future-Gain is unclear.
>
> **A7.** FG-TTL processes sequence boundaries by applying replicate padding for local volatility and strict exclusion for Future-Gain computation:
> * **Boundary computation.** For local volatility calculation, boundary entropy values are duplicated to maintain continuous variance. For Future-Gain, any token lacking a complete adaptive window is excluded from the update pool to prevent asymmetric estimates of uncertainty reduction.
> * **Reason for exclusion.** Early tokens are prompt-dominated and lack enough generated context, so exclusion avoids biased gain estimates while preserving their conditioning role.
>
> >Q8. The role of $T_t$ in backpropagation is unclear.
>
> **A8.** Eq. 11 fully replaces Eq. 4 during backpropagation, and all updates are computed from Eq. 11. The dynamic temperature $T_t$ is detached from the computational graph and only used to modulate gradient strength.
>
> >Q9. Comparison with ZeroSiam is unclear.
>
> **A9.** **FG-TTL and structural regularization are complementary**: FG-TTL filters harmful gradients through future-aware token selection, while ZeroSiam prevents collapse through asymmetric structural regularization. Structural alignment may further stabilize adaptation and reduce sensitivity to heuristic thresholds.

---

> > ### Author Rebuttal · Reviewer_3YB7 · 2026-04-01
> >
> > Thank you for the rebuttal. The added clarifications are helpful, but they do not fully resolve my main concerns. In particular, the response mostly restates the method as future-aware, risk-aware, and source-free, but it still does not convincingly establish conceptual novelty beyond a careful combination of selective entropy-based updating and uncertainty-aware gradient modulation. More importantly, the theoretical grounding of Future-Gain remains insufficient: the paper provides a rigorous analysis for temperature-based gradient scaling, but still lacks a principled argument for why the specific Future-Gain formulation in Eq. (7) should track token informativeness or performance improvement, rather than functioning as an empirical heuristic. The extra sensitivity and decoding results are useful, but they do not fully address the ad hoc nature of the selection protocol or the gap between the narrative and the actual role of the entropy threshold.
> >
> > ---
> >
> > *Additional note*: Thank you for the further reply. The added clarification is helpful, especially the CMI-based interpretation of Future-Gain, but it still does not resolve my main concerns. I remain unconvinced that the method establishes conceptual novelty beyond a careful combination of selective entropy-based updating and uncertainty-aware gradient modulation, and I still do not find Eq. (7) sufficiently principled rather than a reasonable empirical surrogate. The added sensitivity results are useful, but they do not fully remove my concern that the selection protocol remains somewhat heuristic.

---

> > > ### Author Response · Authors · 2026-04-02
> > >
> > > Thanks for your follow-up and for raising your score to 4.
> > > >Q1. Novelty of the FG-TTL.
> > >
> > > **A1.** Key novelty of FG-TTL lies in **redefining token update utility for autoregressive TTL.** Instead of using instantaneous uncertainty as the signal of what to update, FG-TTL prioritizes tokens by their contribution to downstream uncertainty reduction. Thus, our contribution is not merely a careful combination of selective updating and gradient control, but a different token-level adaptation principle for autoregressive TTL.
> > > * **Existing entropy-based TTA/TTL methods do not explicitly define token update utility for autoregressive generation.** Prior entropy-based TTA is mainly sample-level, and recent TTL methods typically rely on instantaneous uncertainty such as entropy or perplexity. In LLMs, however, a token’s update value depends not only on its current uncertainty, but also on whether it improves the downstream decoding trajectory.
> > > * **Core novelty is a future-oriented token utility criterion based on downstream uncertainty reduction.** Future-Gain is not simply another score for selecting hard tokens by instantaneous entropy. Instead, it identifies tokens that act as local pivots in generation, i.e., tokens whose realization is followed by reduced downstream predictive uncertainty. Therefore, the conceptual change introduced by FG-TTL is from selecting tokens that are **currently uncertain** to selecting tokens that **are worth learning from for future generation**. This interpretation is also consistent with our empirical results.
> > > * **FTS and RAA form a unified design around the same principle.** FTS decides where to learn via Future-Gain, while RAA decides how strongly to learn by suppressing unreliable gradients on the selected tokens. Thus, the contribution is a unified autoregressive adaptation mechanism rather than two loosely combined modules.  Our ablation results are consistent with this interpretation.
> > > * **FG-TTL differs from recent methods in formulation and adaptation signal.** Recent LLM methods explore perplexity-based TTL, joint uncertainty adaptation, or RL-style adaptation on unlabeled data. FG-TTL instead focuses on token-level autoregressive TTL driven by intrinsic uncertainty signals from the unlabeled test stream.
> > >
> > >
> > > >Q2. Theory of Future-Gain.
> > >
> > > **A2:** We provide a formal derivation showing that Future-Gain is a computationally tractable surrogate for the Conditional Mutual Information (Bialek et al., 2001) that the current token $y_t$ carries about the future trajectory.
> > >
> > > **Information-Theoretic Objective:** In autoregressive generation, the reduction in future uncertainty upon generating $y_t$ is quantified by CMI. $Y_{t+1:t+N}$ as the future trajectory of window size $N$ and $Y_{<t}$ as the historical context, the CMI is:
> > > $$I(Y_{t+1:t+N} ; y_t \mid Y_{<t}) = H(Y_{t+1:t+N} \mid Y_{<t}) - H(Y_{t+1:t+N} \mid Y_{\le t})$$
> > > where $Y_{\le t} = Y_{<t} \cup \{y_t\}$.
> > >
> > > **Posterior Entropy Expansion:** By the chain rule of Shannon entropy:
> > > $$H(Y_{t+1:t+N} \mid Y_{\le t}) = \sum_{j=1}^{N} H(y_{t+j} \mid Y_{<t+j}) := \sum_{j=1}^{N} H_{t+j}$$
> > > Averaging this sum corresponds exactly to $H_{future}$ in our method.
> > >
> > > **Local Stationarity Approximation:** The first term—the prior uncertainty before $y_t$ is determined—is intractable to compute online, requiring vocabulary-wide marginalization and future tree search. We address this via a **Local Stationarity** assumption: a model's intrinsic uncertainty remains locally stable within a reasoning chain unless an information pivot occurs. Thus, we proxy this unobservable prior using the average entropy of the immediate past $N$ steps ($H_{past}$).
> > >
> > > **Deriving Future-Gain:** Substituting these components into the normalized CMI transforms it into an empirical difference measurable along a single trajectory:
> > > $$
> > > \frac{1}{N} I(Y_{t+1:t+N} ; y_t \mid Y_{<t}) \approx \frac{1}{N} \sum_{i=1}^{N} H_{t-i} - \frac{1}{N} \sum_{j=1}^{N} H_{t+j} =\mathcal{G}_t
> > > $$
> > >
> > > Here, the first term is the prior proxy $H_{\mathrm{past}}$, and the second term is the realized posterior $H_{\mathrm{future}}$. This is matches Future-Gain in Eq. 7.  With adaptive window $N(t)$ to account for varying information propagation ranges, Future-Gain operates as a principled, computationally feasible surrogate for CMI rather than an ad-hoc heuristic. We will include this derivation in the revised version.
> > >
> > >
> > > >Q3. Selection protocol and the role of the entropy threshold.
> > >
> > > **A3.** The protocol is not ad hoc because $\xi_{low}$ and Future-Gain serve distinct roles: $\xi_{low}$ only filters trivial near-deterministic tokens, while Future-Gain performs the actual ranking by downstream uncertainty reduction. Thus, the threshold is a validity filter rather than the pivot-defining criterion, and the improvement comes from Future-Gain ranking, not entropy sparsification. The weak sensitivity to $\xi_{low}$ further supports that the gain is driven by Future-Gain rather than a particular threshold choice.

---

### Official Review · Reviewer_FMKo · 2026-03-09

**Soundness:** 1
**Presentation:** 2
**Significance:** 2
**Originality:** 2
**Overall Recommendation:** 2
**Confidence:** 4

**Summary:**

FG-TTL is an online test-time learning method for LLMs that updates LoRA parameters on unlabeled test streams. The core idea is to adapt only on “informative” generated tokens: the method defines Future-Gain to select tokens that appear to reduce downstream uncertainty, then uses Risk-Aware Adaptation to scale updates by gain and entropy-aware temperature. Experiments on six reasoning benchmarks, plus DomainBench and a quantized setting, show average improvements over several TTL/TTA baselines.

**Compliance With Llm Reviewing Policy:**

Affirmed.

**Final Justification:**

While the rebuttal adds some aggregate efficiency numbers and limited before/after evidence, it does not resolve the main concerns: the adaptation signal remains self-referential and correctness-agnostic, the method’s compatibility with realistic LLM serving remains unclear, and the paper still lacks convincing evidence of robustness under long, non-stationary online streams. I therefore remain unconvinced about both the practical value and the technical soundness of the proposed method.

**Key Questions For Authors:**

- What is the actual serving-time overhead of FG-TTL, including backward pass cost, memory footprint, and interaction with continuous batching / KV-cache-based inference systems? This is currently missing and is central to judging practical value.

- How stable is the method over long, uncontrolled streams? In particular, do the authors have evidence that FG-TTL does not drift, overfit recent traffic, or catastrophically forget under adversarial or highly imbalanced inputs?

- How stable is the method over long, uncontrolled streams? In particular, do the authors have evidence that FG-TTL does not drift, overfit recent traffic, or catastrophically forget under adversarial or highly imbalanced inputs?

**Strengths And Weaknesses:**

### Strengths:
- The paper tackles a timely problem: adapting LLMs under distribution shift without labels or retrieval access, and the motivation for token-selective adaptation in autoregressive decoding is reasonable.

- Multiple backbones, six reasoning datasets, component ablations, strategy comparisons, quantized evaluation, and DomainBench. The gains are not huge, but they are generally consistent.

### Weaknesses:
- The method still requires online backpropagation through LoRA during inference, which is hard to reconcile with production LLM serving stacks built around aggressive forward-only optimization, batching, and KV-cache efficiency. The paper does not seriously address latency, throughput, systems complexity, or serving compatibility. I would like to ask the authors to talk something about this point, it is a practical deployability issue.

- The paper argues that RAA suppresses collapse and noisy gradients, but the evidence is mostly indirect. There is no convincing long-horizon streaming analysis showing robustness under adversarial, highly skewed, or non-stationary user traffic, which is exactly where online self-updating methods are most risky

-  Future-Gain is defined from local entropy differences over an adaptive window, but it is still a proxy rather than a principled estimator of beneficial learning signal.

- Even though marginal improvements exist, they are not strong enough to outweigh the engineering and reliability cost of doing parameter updates at test time.

- In practice, distribution shift is often handled by retrieval/context injection or by improving offline training / test-time compute, rather than by mutating model parameters on live traffic.

---

> ### Author Rebuttal · Authors · 2026-03-31
>
> We are deeply grateful to you for recognizing the strengths of our work, particularly the " **motivation ... is reasonable**", and "**gains are generally consistent**".
>
> >Q1. Latency, throughput, and serving compatibility are underexplored.
>
> **A1.** We added efficiency analysis of FG-TTL and all baselines (see Tab. A).
> * **Moderate overhead.** FG-TTL does introduce additional cost, but the latency increase is moderate. FG-TTL takes 39.30s total latency, which is comparable to EATA and lower than TLM.
> * **Favorable efficiency–performance trade-off.** FG-TTL  achieves the best overall trade-off between performance and efficiency: it delivers the highest accuracy together with the highest throughput, while maintaining competitive additional memory usage.
> * **Practical relevance.** FG-TTL adds only a moderate runtime cost in practice, while the resulting accuracy gain makes FG-TTL promising for deployment scenarios where both adaptation quality and efficiency are important.
>
> Tab A: Efficiency and Performance on MATH500 under Llama3.1-8B
> |Method|Total Latency (s)|Extra Mem (GB)|Throughput (tokens/s)|Acc|
> |-|-|-|-|-|
> |Tent|40.42|14.31|31.58|49.20|
> |EATA|38.43|13.98|31.97|49.40|
> |COME|**37.27**|23.72|31.05|48.80|
> |TLM|42.30| **0.78**|31.61|50.00|
> |Ours|39.30|13.61|**32.01**|**51.60**|
>
> >Q2. Robustness under adversarial and non-stationary streams is unclear.
>
> **A2.** Thanks for your insightful comment. **Please see Reviewer-EbpV, A6.**
>
> >Q3. Future-Gain is a proxy, not a principled estimator.
>
> **A3.** In a strictly unsupervised online setting, where GT is unavailable, any criterion for guiding updates is necessarily a proxy rather than a perfect estimator. Our claim is that Future-Gain is a more theoretically motivated and empirically effective proxy than instantaneous entropy.
> * **Theory-motivated design.** Future-Gain is inspired by predictive information and information-bottleneck intuitions: instead of measuring only the confidence of the current token, it evaluates whether the current token helps reduce downstream uncertainty in autoregressive generation. This makes it better aligned with beneficial learning signal than purely local confidence.
> * **Empirical support.**  From Tab. 3 and Fig. 2c, Future-Gain provides more stable and effective updates than instantaneous entropy, leading to stronger performance across multiple reasoning tasks.
>
>
> >Q4. Gains may not justify TTU.
>
> **A4.** The gains of FG-TTL are meaningful and come with relatively low adaptation overhead:
> * **Lightweight Updates.** FG-TTL is intentionally designed to be lightweight. It updates only LoRA and runs in an online small-batch setting without requiring external retrieval systems.
> * **Meaningful Gains in a Challenging Setting.** TTL setting is inherently difficult: adaptation must be performed without labels. In this setting, even moderate gains are meaningful. Notably, the improvements are not merely marginal on challenging benchmarks. For example, on AIME24, FG-TTL improves performance **from 3.33% to 13.33%**.
>
> >Q5. Why use TTL instead of retrieval or offline scaling?
>
> **A5.**  Our goal is not to replace these paradigms, but to **study a complementary source-free online TTL setting** where external data, and retrieval infrastructure may be unavailable.
> * **Different deployment setting.** RAG and TTT methods typically assume access to external corpora, or source data, while offline training addresses shifts only before deployment. In contrast, **our work focuses on the setting where the model must adapt using only the incoming unlabeled test stream, without external data**.
> * **Data Dependency and Problem Settings.** FG-TTL does not perform full model updates on live traffic. Instead, it updates only LoRA, **without requiring retrieval systems or multiple expensive decoding passes**.
> * **Continual adaptation benefits.**  Test-time compute mainly explores more trajectories under the current model. In contrast, TTL **provides a persistent and cumulative mechanism for aligning the model to recurring patterns in the test stream**.
>
> >Q6. Risk of overfitting, and forgetting is unclear.
>
> **A6.** FG-TTL exhibits stable adaptation behavior with little evidence of forgetting:
> * **Experimental setup.** We first evaluate the original LLM on agriculture_5k to obtain the reference (“Before”). We then adapt the model separately on each reasoning stream, and evaluate each adapted model again on agriculture_5k (“After”).
> * **FG-TTL shows strong resistance to forgetting.** From Tab. B, the performance on agriculture_5k remains highly stable after adaptation on the reasoning datasets: the average score stays unchanged, and the variations are all very small.
>
> Tab. B: Result on agriculture_5k before and after TTL on mathematical datasets
> | |GSM8K|MATH500|College|AIME24|Minerva|Olympiad|Avg|
> |-|-|-|-|-|-|-|-|
> |Before|8.76|8.76|8.76|8.76|8.76|8.76|8.76|
> |After|8.71|8.85|8.79|8.75|8.87|8.56|8.76|
> |$\Delta$|-0.05|+0.09|+0.03|-0.01|+0.11|-0.20|0.00|

---

> > ### Author Rebuttal · Reviewer_FMKo · 2026-04-02
> >
> > While the rebuttal adds some aggregate efficiency numbers and limited before/after evidence, it does not resolve the main concerns: the adaptation signal remains self-referential and correctness-agnostic, the method’s compatibility with realistic LLM serving remains unclear, and the paper still lacks convincing evidence of robustness under long, non-stationary online streams. I therefore remain unconvinced about both the practical value and the technical soundness of the proposed method.

---

> > > ### Author Response · Authors · 2026-04-05
> > >
> > > >Q1. The adaptation signal remains self-referential and correctness-agnostic.
> > >
> > > **A1.** We thank the reviewer for this important concern. We respond in three points:
> > > * **What the question is in TTL.** In source-free, unlabeled TTL, token-level correctness is not observable at deployment time. The key question is therefore not direct correctness estimation, but which intrinsic signals better identify useful online updates. Our method is designed for this question.
> > > * **What Future-Gain captures.** Future-Gain does not reward a token simply for being confident itself. Instead, it measures whether a token is followed by reduced downstream predictive uncertainty. Thus, while the signal is intrinsic to the model, it is **not merely token-self-confirmatory**; it captures whether a token is associated with a clearer subsequent trajectory.
> > > * **Why it is useful.** Prior work [r1] suggests that model uncertainty is often informative of correctness/reliability, though imperfectly. Motivated by this, we use reduced downstream uncertainty as a practical signal that a token lies on a more reliable decoding trajectory, and is thus more useful for online adaptation than instantaneous entropy. This is also consistent with our results. Therefore, our claim is not that Future-Gain directly estimates correctness, but that it provides a better intrinsic proxy for token update utility than instantaneous entropy in source-free TTL.
> > >
> > > [r1] Detecting hallucinations in large language models using semantic entropy, Nature.
> > >
> > > Tab A: Comparison of selection strategies
> > > |Strategy|GSM8K|MATH500|CollegeMath|AIME24|Avg|
> > > |-|-|-|-|-|-|
> > > |LLM|82.18|49.20|25.00|3.33|39.93|
> > > |Random|80.44|49.33|25.39|10.00|41.29|
> > > |High Ent.|80.06|50.04|24.83|6.67|40.40|
> > > |Ours|**83.40**|**51.60**|**25.92**|**13.33**|**43.56**|
> > >
> > > >Q2. The method’s compatibility with realistic LLM serving stacks remains unclear.
> > >
> > > **A2.** FG-TTL targets distinct, highly practical use cases where adaptability is prioritized, and it can be engineered to fit high-throughput systems seamlessly.
> > > * **Targeted deployment scenarios.** TTL is fundamentally designed for environments where continuous adaptation to non-stationary or private data outweighs the need for massive concurrent throughput. Highly relevant scenarios include on-device personalized agents, autonomous agents navigating novel environments, and offline enterprise batch-processing pipelines (e.g., medical). In these settings, the moderate overhead of LoRA updates is highly acceptable and practically deployable.
> > > * **Asynchronous shadow-updating for high-throughput systems.** Even in high-concurrency serving stacks, backward passes need not block inference. FG-TTL can adopt an asynchronous design: the main engine remains forward-only to preserve KV-cache and continuous batching, while a background process computes gradients only on selected high-gain tokens and periodically syncs the lightweight LoRA parameters back to the serving engine (e.g., via dynamic LoRA loading frameworks such as S-LoRA).
> > >
> > >
> > > >Q3. Lack of convincing evidence regarding robustness under long, non-stationary, or adversarial online streams.
> > >
> > > **A3.** To directly address this concern, we add two stress tests: (1) a noisy mixed-stream test to simulate hostile/corrupted traffic, and (2) a long-horizon continual adaptation test across heterogeneous domains.
> > > * **Robustness under noisy/adversarial streams.** To simulate real-world hostile traffic, we corrupted 20% of the MATH500 test samples by severely perturbing 30% of their words (random deletion, swapping, or ASCII string replacement). In Tab. B, FG-TTL performs best, suggesting stronger robustness to noisy online updates.
> > > * **Stability under long-horizon non-stationary continual adaptation.** We continually adapt without resetting parameters on GSM8K→MATH500→AIME24→Minerva. In Tab. C, continual adaptation remains close to isolated adaptation, indicating that FG-TTL stays stable under multi-stage distribution shifts rather than drifting or collapsing.
> > > * **Minimal catastrophic forgetting.** To test if live updates wipe out pre-trained knowledge, we evaluated the continually adapted model on a holdout domain (agriculture_5k) at each step. In Tab. D, the performance remains highly stable throughout the entire multi-stage adaptation process. FG-TTL effectively absorbs new domain patterns without overwriting its foundational knowledge base.
> > >
> > > Tab. B: Robustness under a noisy mixed-stream stress test on MATH500
> > > |Method|Acc|
> > > |-|-|
> > > |Base|44.80|
> > > |Tent|43.00|
> > > |EATA|42.80|
> > > |COME|44.00|
> > > |TLM|42.20|
> > > |Ours|**45.60**|
> > >
> > > Tab. C: Isolated vs. Continual Learning on target tasks
> > > |Method|GSM8K|MATH500|AIME24|Minerva|
> > > |-|-|-|-|-|
> > > |Base Model|82.18|49.20|3.33|20.96|
> > > |Isolated Adaptation|83.40|51.60|13.33|22.79|
> > > |Continual Adaptation|83.40|50.80|10.00|23.16|
> > >
> > > Tab. D: Holdout base knowledge (Agri) during continual Learning
> > > |Step|Base (Start)|After GSM8K|After MATH500|After AIME24 |After Minerva|
> > > |-|-|-|-|-|-|
> > > |Agri|8.76|8.71|8.43|8.41|8.45|

---

### Official Review · Reviewer_EbpV · 2026-03-12

**Soundness:** 4
**Presentation:** 4
**Significance:** 4
**Originality:** 4
**Overall Recommendation:** 5
**Confidence:** 4

**Summary:**

This paper addresses the challenge of adapting Large Language Models (LLMs) to distribution shifts during inference using unlabeled test data. The authors identify two primary failures in existing test-time learning (TTL) methods based on entropy minimization: temporal error propagation and model collapse due to updates on unreliable tokens. To solve this, they propose Future-Gain Guided Test-Time Learning (FG-TTL), which focuses on updating the model only on tokens that reduce uncertainty in subsequent generations rather than tokens that are simply uncertain at the current step. The framework consists of Future-Gain Guided Token Selection (FTS) to pinpoint informative positions and a Risk-Aware Adaptation (RAA) mechanism that combines gain-based weighting with adaptive temperature scaling to suppress unreliable gradients. The method is evaluated across 6 benchmarks with Llama and Qwen models, demonstrating superior average performance.

**Compliance With Llm Reviewing Policy:**

Affirmed.

**Final Justification:**

The rebuttal addressed my concerns, and I'm maintaining my recommendation of Accept. The added efficiency analysis confirms the latency overhead is reasonable, while the new stress tests and open-ended generation results prove robustness and generalizability. While other reviewers raise valid concerns about the engineering hurdles of online LoRA updates in production serving stacks, I believe this paper stands as a strong algorithmic contribution. The shift from relying on entropy to optimizing for future gain is well-motivated.

**Key Questions For Authors:**

1. Computing the Future-Gain metric requires evaluating subsequent entropies via a forward lookahead window, either fixed or adaptive. Could you quantify the exact wall-clock inference latency overhead this adds compared to baseline TTL methods?
2. The Risk-Aware Adaptation relies on the parameter $\alpha$ to dynamically modulate the temperature $T_{t}$ based on instantaneous entropy. How sensitive is the model's overall stability and performance to the selection of this specific parameter?
3. The method is thoroughly evaluated on complex mathematical and scientific reasoning benchmarks. How does FG-TTL perform on highly open-ended creative generation tasks where the concept of a "correct" trajectory is significantly less constrained?

**Limitations:**

The authors discuss the risk of updating on unreliable tokens and suggest the RAA mechanism as a primary defense. However, a more explicit discussion of the method's behavior when faced with "black swan" distribution shifts (data that is completely outside the model's domain or structural capability) would be beneficial.

**Strengths And Weaknesses:**

Strengths

Soundness: The paper provides a strong technical foundation, including a theoretical derivation for gradient scaling via temperature adjustment and a detailed analysis of "hidden pivots" in the vocabulary.

Presentation: The paper is exceptionally clear and well-structured. The logical flow from initial observations to the proposed submodules (FTS and RAA) is supported by informative visualizations like Figure 1.

Significance: Achieving a 1.6% improvement over the strongest baseline on Llama 3.1-8B and showing stability on high-difficulty benchmarks like AIME24 indicates high practical utility for deploying LLMs in evolving real-world environments.

Originality: The shift from instantaneous entropy to "Future-Gain" is a well-motivated departure from standard TTL methods. By targeting "informational pivots" the approach effectively addresses the unique temporal coupling of autoregressive generative models.

Weaknesses

Soundness: While the authors note the method is efficient by updating only LoRA parameters, the requirement to compute future entropies (via the Future Window) implies a potential latency increase during inference. The paper would benefit from a more detailed quantification of wall-clock time overhead.

Presentation: While the comparison includes SOTA TTL and TTA methods, a direct comparison with the most recent retrieval-augmented test-time training (TTT) methods is absent, which would provide a broader view of the current landscape.

Significance: The system introduces new parameters such as the adaptive window $N(t)$ and the filtering threshold $\tau$. Although FTS aims to reduce manual tuning by dynamically adjusting the window size, the sensitivity of the overall adaptation performance to these initial bounds and the specific threshold $\tau$ is not extensively explored in the current results. Exploring these sensitivities would be important for understanding the robustness of the method across different hardware constraints and real-time latency requirements.

---

> ### Author Rebuttal · Authors · 2026-03-31
>
> We are deeply grateful to you for recognizing the strengths of our work, particularly the "strong technical foundation", "**well-motivated**", and "**high practical utility**".
>
> >Q1. Wall-clock overhead is underexplored.
>
> **A1.** We added efficiency analysis of FG-TTL and all baselines (see Tab. A).
> * **Moderate overhead.** FG-TTL does introduce additional cost, but the latency increase is moderate. FG-TTL takes 39.30s total latency, which is comparable to EATA (38.43s) and lower than TLM (42.30s).
> * **Favorable efficiency–performance trade-off.** FG-TTL achieves the best overall trade-off between performance and efficiency: it delivers the highest accuracy together with the highest throughput, while maintaining competitive additional memory usage.
>
> Tab A: Efficiency and Performance on MATH500 under Llama3.1-8B
> |Method|Total Latency (s)|Extra Mem (GB)|Throughput (tokens/s)|Acc|
> |-|-|-|-|-|
> |Tent|40.42|14.31|31.58|49.20|
> |EATA|38.43|13.98|31.97|49.40|
> |COME|**37.27**|23.72|31.05|48.80|
> |TLM|42.30| **0.78**|31.61|50.00|
> |Ours|39.30|13.61|**32.01**|**51.60**|
>
> >Q2. Missing comparison with TTT methods.
>
> **A2.** We would like to clarify the rationale for not including TTT methods, along with the potential advantages of FG-TTL:
> * **External data and retrieval dependency.** **TTT methods** typically **rely on a well-maintained retrieval corpus or source data** to fetch supportive examples for adaptation. In contrast, our method deliberately assumes a stricter source-free setting and adapts using only the test data itself.
> * **Experimental design principle.** Our experimental protocol is aligned with the TTL setting, where no labels or external knowledge sources are accessible at test time. Therefore, we compare against TTA  and TTL, which operate under the same or closely related assumptions. While TTT methods are powerful in their own setting, they fall outside the scope of our source-free evaluation due to their reliance on retrieval infrastructure.
>
> >Q3. Sensitivity to $\tau$ is unclear.
>
> **A3.** We would like to clarify as:
> * **Clarification of notation.** FG-TTL does not include threshold $\tau$. We believe you are referring to the entropy filtering threshold $\xi_{low}$ used in FTS.
> * **Sensitivity analysis.** We added a sensitivity study of $\xi_{low}$. Tab. B shows that performance is generally stable across this range, with the best result achieved at $\xi_{low}=10^{-4}$. This suggests that FG-TTL is not overly sensitive to $\xi_{low}$ and does not require careful threshold tuning.
>
> Tab. B: Impact of $\xi_{low}$
> |$\xi_{low}$|1e-3|1e-4|1e-5|
> |-|-|-|-|
> |GSM8K|83.09|**83.40**|82.94|
> |MATH500|50.20|**51.60**|50.00|
>
> >Q4. Sensitivity to $\alpha$ is unclear.
>
> **A4.** We added a sensitivity analysis, varying $\alpha \in \{0.1, 0.2, 0.5, 0.7, 1.0\}$. From Tab. C, performance remains stable across this range, with the best result at $\alpha=0.5$ and only minor variation for other values.
>
> Tab. C: Impact of $\alpha$
> |$\alpha$|0.1|0.2|0.5|0.7|1.0|
> |:-| :-| :-| :-| :- |:-|
> |GSM8K|82.26|83.02|**83.40**|82.49|82.79
> |MATH500|50.20|50.00|**51.60**|49.80|51.00|
>
> >Q5. Performance on creative generation is unclear.
>
> **A5.** FG-TTL is not limited to constrained reasoning and remains effective on open-ended generation tasks：
> * **Not Limited to Reasoning.** We evaluate FG-TTL on DomainBench, which covers vertical-domain QA tasks. From Table 5, FG-TTL achieves the best average performance.
> * **New Open-Ended Results.** We evaluate FG-TTL on WritingPrompts using 100 randomly sampled examples. From Tab. D, FG-TTL achieves the best performance.
> * **Method Applicability.** FG-TTL does not rely on a single “correct” generation trajectory. Instead, it selectively updates on tokens that help reduce downstream uncertainty and improve the ongoing decoding process.
>
> Tab D: Result on Creative Generation (WritingPrompts)
> |Method|R-L|BS|Avg|
> |-|-|-|-|
> |Base|0.1370|0.4954|0.3162|
> |Tent|0.1357|0.4943|0.3150|
> |EATA|0.1360|0.4947|0.3154|
> |COME|0.1354|0.4930|0.3142|
> |TLM|0.1363|0.4946|0.3155|
> |Ours|**0.1375**|**0.4960**|**0.3168**|
>
> >Q6. Robustness to black-swan shifts is unclear.
>
> **A6.** We added the following stress test:
> * **Stress-test setup.** We conduct a noisy mixed-stream experiment on MATH500. Specifically, 20% of the test samples are randomly selected as noisy inputs. For each selected noisy sample, 30% of the words in its text field are corrupted with severe word-level perturbations, including random deletion, adjacent-word swapping, or replacement with random ASCII strings of the same length.
> * **FG-TTL remains more robust.** From Tab. E, under this hostile setting, the TLM drops to 42.2%, while FG-TTL reaches 45.6%. This indicates that FG-TTL remains more robust than the standard TTL baseline even under severe and partially adversarial corruption.
>
> Tab E: Robustness under a noisy mixed-stream stress test
> |Method|Acc|
> |-|-|
> |Base|44.80|
> |Tent|43.00|
> |EATA|42.80|
> |COME|44.00|
> |TLM|42.20|
> |Ours|**45.60**|

---

> > ### Author Rebuttal · Reviewer_EbpV · 2026-04-02
> >
> > Thank you to the authors for the thorough rebuttal and for running the additional experiments. I appreciate the clear breakdown of the wall clock latency, as well as the new results on open-ended generation and the stress tests.
> >
> > After reviewing your responses, my initial questions and concerns have been adequately addressed:
> >
> > Latency and Overhead: The efficiency analysis provided in Table A clarifies the operational cost. A total latency of 39.30s is a reasonable overhead for this setting and demonstrates a favorable performance-efficiency trade-off compared to the EATA and TLM baselines.
> >
> > Robustness and Open-Ended Tasks: The stress test showing 45.6% accuracy under severe word-level corruption builds confidence in the Risk-Aware Adaptation mechanism's ability to handle noisy shifts. Additionally, the results on WritingPrompts prove the method's viability beyond strictly constrained mathematical reasoning tasks.
> >
> > Parameter Sensitivity: The ablation on the temperature scaling parameter $\alpha$ confirms that the method is relatively stable across a sensible range and does not require brittle, dataset-specific tuning.
> >
> > While other reviewers have rightly pointed out the practical engineering hurdles of deploying online LoRA updates within highly optimized, production serving stacks, I view this work as a valuable algorithmic contribution to the specific paradigm of test-time learning. The conceptual shift from relying on instantaneous entropy to optimizing for downstream Future-Gain is well-motivated and conceptually sound.
> >
> > I am keeping my overall recommendation at a 5 (Accept).

---

> > > ### Author Response · Authors · 2026-04-05
> > >
> > > Thank you very much for your careful follow-up and thoughtful acknowledgment. We greatly appreciate your positive assessment of our rebuttal and the additional experiments. We are glad that our clarifications on efficiency, robustness, and stability helped address your concerns, and we sincerely thank you for your support of this work.

---

### Decision · Program_Chairs · 2026-04-30

**Decision:**

Accept (regular)

**Comment:**

The submission "Future-Gain Guided Test-Time Learning for Large Language Models" describes a test-time adaptation strategy for large language models that aims to identify pivot tokens in sequences self-generated by the model without supervision and on which the model can be updated to reduce uncertainty. The authors show that the proposed strategy improves performance on six benchmarks for several  language models.

Reviewers agree that the problem studied in the submission is very timely, and the submission is overall structured well and clearly explained.

Reviewers are however, polarized regarding the merits of the proposed approach, with concerns along several lines:
* Whether the improvements shown are significant enough
* How improvements relate to the complexity and motivation of the approach and whether the approach is overall more akin to  'a careful combination of selective entropy-based updating and uncertainty-aware gradient modulation than to a fundamentally new adaptation principle' (reviewer 3YB7)
* Whether the method (which requires online updates) is reconcilable with current-day production environments of LLMs (and whether this is a lens through which TTL methods should be evaluated)
* Whether the approach would work in practical deployment where there may be noise, non-stationary traffic or other effects that would break the self-update rules proposed in the submission.
* How principled the framing as "future gain guided learning" is.


Overall, I do think the production concerns (while crucial for current-generation deployment) are forgiveable for a submission to ICML about future online learning techniques, and I am arguing that the approach is interesting enough to be discussed at the conference.